# Conserved and divergent features of human mRNA decapping revealed by biochemical reconstitution

Eric A. J. Simko[1,3], Sowndarya Muthukumar [2,3] ✉, Tanner M. Myers[1], Anna L. Valkov[1] & Eugene Valkov [1] ✉

Decapping is a critical step in mRNA decay, but the mechanisms regulating human decapping enzyme DCP2 remain poorly understood. Here, we reconstitute the human decapping network using full-length recombinant proteins and compare it to the yeast system. Unlike in yeast, we find that the C-terminal region of human DCP2 is not autoinhibitory. RNA-binding residues of yeast Dcp2 are not conserved in the human homolog, and we find instead that a charged C-terminal region mediates substrate recognition. Human DCP1 does not stably interact with or directly stimulate DCP2, but mediates activation by the enhancer PNRC2. We also demonstrate that decapping enhancer EDC4 forms tetramers through an extended coiled-coil region, and that both DCP1 and EDC4 homomeric species can further assemble into higher-order oligomers. Furthermore, structural predictions incorporating these findings suggest a model for DCP2 recruitment by EDC4 tetramers. These findings reveal key mechanistic differences between human and yeast decapping regulation and provide insight into the molecular architecture underlying mRNA decay.

Gene expression levels depend on a careful balance between mRNA synthesis and decay. Steps in mRNA degradation provide critical regulatory points in cell division[1] and differentiation[2] and allow for dynamic responses to external stimuli[3,4] and viral infection[5]. Removing the 5′ cap structure, or decapping, is a decisive and penultimate step in the 5′-to-3′ mRNA decay pathway. Decapping commits transcripts to rapid degradation by the exoribonuclease XRN1[6,7]. In eukaryotes, this decapping step is carried out by the Nudix hydrolase DCP2[8–10]. Unlike many cap-binding factors, which directly recognize the 5′ cap structure, such as eIF4E[11], DCP2 relies on interactions with the RNA body rather than the cap itself for substrate recognition[9,10].

Given the decisive nature of decapping in mRNA decay, it is a tightly regulated step. A network of decapping enhancers mediates this regulation through direct modulation of DCP2 catalytic activity and recruits it to target transcripts[12–15]. A key component of this enhancer network is DCP1, which interacts tightly with DCP2 in fission yeast and mediates stimulation by enhancers such as Edc1 and Edc2[12,16]. Yeast Dcp1 consists solely of a structured EVH1 domain, which binds stably to the N-terminal domain of Dcp2 and interacts with decapping enhancers such as Edc1[12,16] (Fig. 1a). Metazoan DCP1, however, consists of a homologous EVH1 domain as well as an extended C-terminal sequence (Fig. 1a). This C-terminal extension is predicted to be largely disordered, except for a helical motif shown to bind the decapping enhancer EDC3 in *Drosophila melanogaster*[14], and a small helical bundle at the C-terminus which is thought to mediate homotrimerization[17]. Evidence suggests that a stable interaction between DCP1 and DCP2 may not be preserved in metazoa, although DCP1 retains a vital role in decapping regulation[16,18]. Current understanding of the DCP1–DCP2 interaction is based largely on studies of colocalization and the activity of immunopurified complexes, which could potentially include unidentified factors.

[1]National Cancer Institute, National Institutes of Health, Frederick, MD, USA. [2]Division of Molecular Hematology, Department of Laboratory Medicine, Lund Stem Cell Center, Faculty of Medicine, Lund University, Lund, Sweden. [3]These authors contributed equally: Eric A. J. Simko, Sowndarya Muthukumar. ✉e-mail: sowndarya.muthukumar@med.lu.se; eugene.valkov@nih.gov

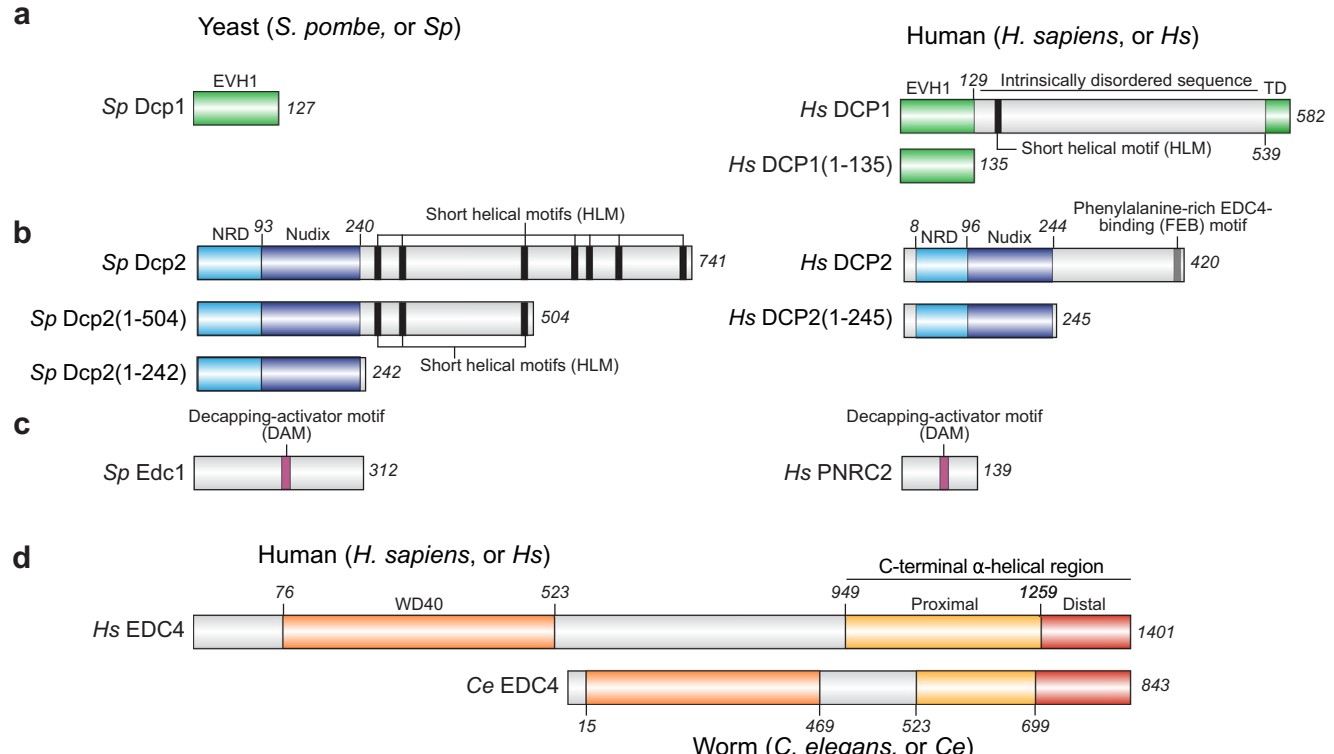

**Fig. 1 | Domain architecture and recombinant constructs of the decapping machinery analyzed in this study. a** *Schizosaccharomyces pombe* Dcp1 (*Sp* Dcp1) consists solely of an EVH1 domain (green). Human DCP1 (*Hs* DCP1) retains this EVH1 domain but is followed by a long intrinsically disordered region (IDR, light gray) that contains a short helical-leucine motif (HLM; black bar) and ends in a trimerization domain (TD; green). *Hs* DCP1(1–135) construct was truncated to include only the N-terminal EVH1 domain. **b** Yeast and human decapping enzymes share conserved N-terminal regulatory (NRD, cyan) and catalytic Nudix (dark blue) domains, followed by species-specific C-terminal IDRs that harbor short linear motifs: multiple HLMs in *Sp* Dcp2 (black bars) and a phenylalanine-rich EDC4-binding (FEB) motif in *Hs* DCP2 (dark gray bar). Constructs include *Sp* Dcp2(1–242)

(structured core only), *Sp* Dcp2(1–504) (core plus partial IDR), full-length *Hs* DCP2 (1–420), and the core construct *Hs* DCP2(1–245). **c** The intrinsically disordered activator *Sp* Edc1 and its functional analog *Hs* PNRC2 each contain a single decapping-activator motif (DAM; magenta bar). **d** Human and *Caenorhabditis elegans* EDC4 share an N-terminal WD40 β-propeller domain (orange) followed by an IDR (gray) and a C-terminal α-helical region that can be divided into proximal (yellow) and distal (red) segments. The nematode protein has a shorter central IDR but preserves the same overall organization. Residue numbers above schematics denote domain boundaries; numbers at the right indicate the final residue (length) of each full-length protein or truncation construct.

While the N-terminal helical and Nudix domains of DCP2 are conserved between yeast and human homologs, the regions C-terminal to the Nudix domain differ significantly (Fig. 1b). Both yeast and human C-terminal segments are predicted to be intrinsically disordered regions (IDRs) containing short functional motifs (Fig. 1b). The C-terminal segment of yeast Dcp2 contains binding elements for decapping enhancers such as Edc3 and Scd6[14] and for substrate targeting via Upf1 and Pat1[15]. While these motifs are not conserved in the human DCP2 ortholog, some of these functions are likely retained by other components of the human decapping network[14,19]. In yeast Dcp2, autoinhibitory motifs have also been identified in the C-terminal IDR[20,21].

Yeast Edc1 and Edc2 are known stimulators of decapping activity[22,23]. Dcp1 mediates this stimulation by binding a proline-rich sequence (PRS) in Edc1/2 through its EVH1 domain[12,24,25]. Dcp1 can also recruit other non-stimulatory decay factors containing similar PRS motifs[24,26]. While there are no known orthologs of Edc1/2 in metazoans, the vertebrate-specific nonsense-mediated decay (NMD) factor PNRC2 (Fig. 1c) contains a PRS that binds yeast and human DCP1 in a manner analogous to the yeast Dcp1:Edc1 interaction[27–29]. PNRC2 is thought to couple NMD and decapping based on the observation that tethering PNRC2 to reporter mRNA induces decay only when PNRC2 is capable of interacting with DCP1[27,28]. Although a truncated mutant of PNRC2 can stimulate the activity of C-terminally truncated DCP2 in the presence of the DCP1 EVH1 domain[28], it is not known whether this synergy occurs with full-length wild-type proteins.

The metazoan-specific decapping enhancer EDC4 is thought to serve as a binding scaffold for DCP2 and decapping regulators[13,30,31] and to facilitate 5′-to-3′ degradation, the final decay step, by recruiting XRN1[18,26,31]. EDC4 is also a critical component of P-bodies, cytoplasmic condensates containing mRNA decay factors as well as translationally repressed transcripts[32]. EDC4 consists of an N-terminal WD40 domain and a C-terminal helical region connected by an extended low complexity region (Fig. 1d). The C-terminal helical region of EDC4 mediates self-association and interactions with XRN1 and DCP2[18,30,33,34]. A phenylalanine-rich EDC4-binding (FEB) motif in DCP2's C-terminal IDR mediates interaction with EDC4[18,34,35], but the mechanism by which EDC4 affects decapping activity remains unknown.

In this work, we investigate the mechanisms of mRNA decapping regulation using a reconstituted system of recombinant purified components to study the eukaryotic decapping network. This system includes several key decapping factors, characterized for the first time in their full-length recombinant forms. Our findings reveal a divergence in the function of the DCP2 C-terminal sequence between yeast and human homologs, with no direct stimulation of DCP2 by DCP1 observed. However, the role of DCP1 as a mediator of enhancer-driven stimulation is conserved, with full-length oligomeric DCP1 facilitating the stimulation of DCP2 by full-length PNRC2. Additionally, we demonstrate that EDC4 forms coiled-coil-based tetramers that can self-associate into megadalton-scale oligomers and provide a model for DCP2 recruitment consistent with the tetrameric state of EDC4.

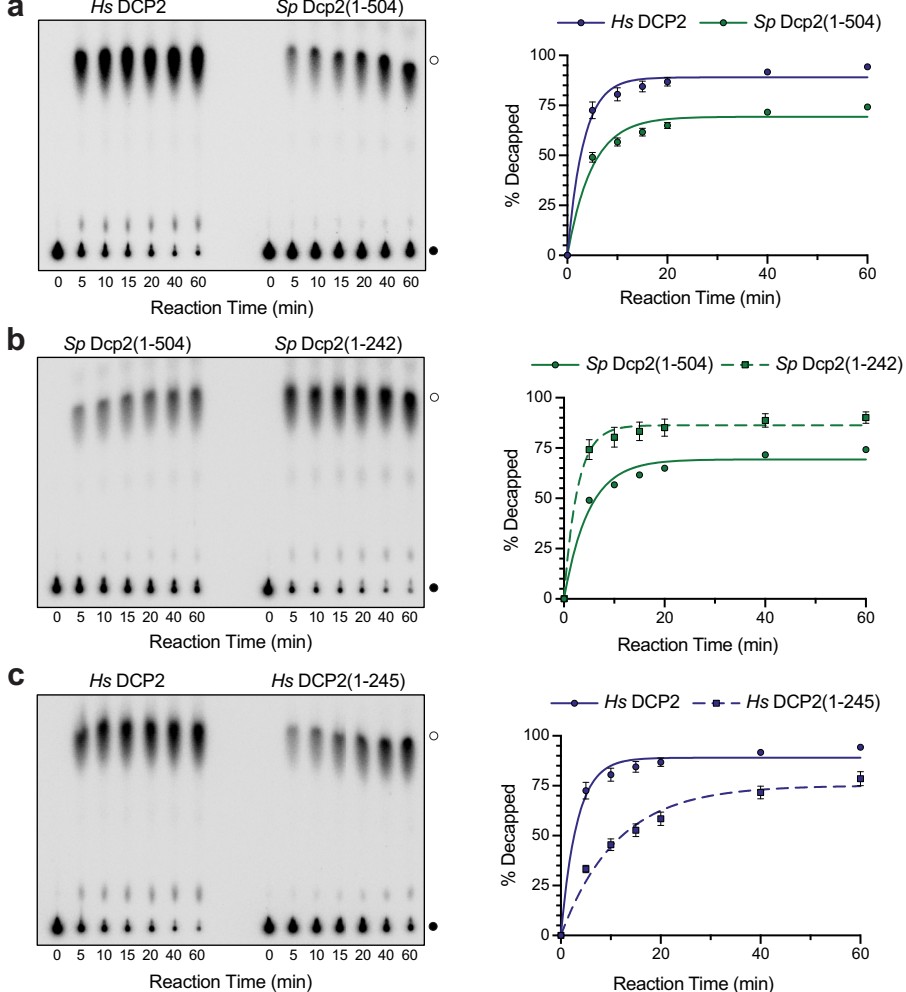

**Fig. 2 | Comparative decapping kinetics of human and yeast DCP2 constructs.**
**a** In vitro decapping assays were performed with full-length *Hs* DCP2 and *Sp*
Dcp2(1–504). Aliquots were quenched at the indicated times (0–60 min), products
and substrate were separated by PEI-cellulose TLC (left), and the percent decapped
RNA was quantified (right). Data points represent the mean ± SEM of three inde-
pendent experiments; error bars that are smaller than the symbols are obscured.
**b** Decapping activity of *Sp* Dcp2(1–504) (structured core + partial IDR) was
compared with the catalytic core *Sp* Dcp2(1–242). Assays and quantification were
carried out as in panel (**a**) (*n* = 3). **c** Full-length *Hs* DCP2 was assayed alongside the
truncated core construct *Hs* DCP2(1–245). Reaction conditions and analysis were
identical to those in panel (**a**) (*n* = 3). In all panels, solid lines show single-
exponential fits to the averaged data, and filled circles mark substrate (●) or
m⁷GDP product (○) positions on TLC plates.

## Results

### Divergent roles of human and yeast DCP2 C-terminal regions

To explore the roles of divergent DCP2 C-terminal sequences, we com-
pared full-length human, *Hs* (*Homo sapiens*), DCP2 with a yeast, *Sp*
(*Schizosaccharomyces pombe*), Dcp2 construct containing known auto-
inhibitory motifs. Because *S. pombe* Dcp2 is considerably larger and
contains extensive low-complexity regions that limit recombinant
expression and sample homogeneity, we performed cross-species
comparisons using a large *S. pombe* Dcp2 fragment encompassing the
conserved catalytic core and a segment of the C-terminal low-complexity
region (residues 1–504), which retains decapping activity in vitro and has
been used previously for biochemical studies[21]; full-length human DCP2
is comparatively tractable and was therefore used throughout. Unlike
previous studies, we directly assessed both homologs under uniform
conditions without decapping enhancers. Decapping activity was mea-
sured using an established in vitro assay that detects the cleavage of
radiolabeled cap from an in vitro transcribed RNA substrate (Supple-
mentary Table 1). We found that human DCP2 exhibited higher basal
activity than the yeast homolog (Fig. 2a). Autoinhibitory motifs in the
yeast Dcp2 C-terminal sequence were previously identified but have only

been studied in the context of copurified Dcp1/2 complexes[21]. To test if
autoinhibition occurs in the absence of Dcp1, we compared the activity
of yeast Dcp2(1–504) with a C-terminally truncated construct,
Dcp2(1–242). Consistent with the Dcp1/2 complex, deleting the yeast
Dcp2 C-terminal region increased activity significantly (Fig. 2b), reaching
levels comparable to full-length human DCP2.

We next tested whether removing the C-terminal region of human
DCP2 would similarly enhance activity. Contrary to expectations,
C-terminal truncation of human DCP2 significantly decreased activity
compared to the full-length enzyme (Fig. 2c). These results indicate a
fundamental divergence in the roles of DCP2 C-terminal regions.

Eukaryotic DCP2 orthologs cannot bind and hydrolyze cap
structures directly and instead rely on recognition of the adjacent RNA
moiety[9,10,36–38]. Substrate binding by yeast Dcp2 has been attributed to
a series of critical lysines on the BoxB helix near the C-terminal end of
the Nudix domain[16,25], but these lysines are not conserved in human
DCP2 (Fig. 3a). Observing that the human C-terminal region is enriched
in positively charged residues (Supplementary Fig. 1a), we wondered
whether the reduced activity of truncated human DCP2 could be a
result of decreased interaction with substrate. To explore the

**a**

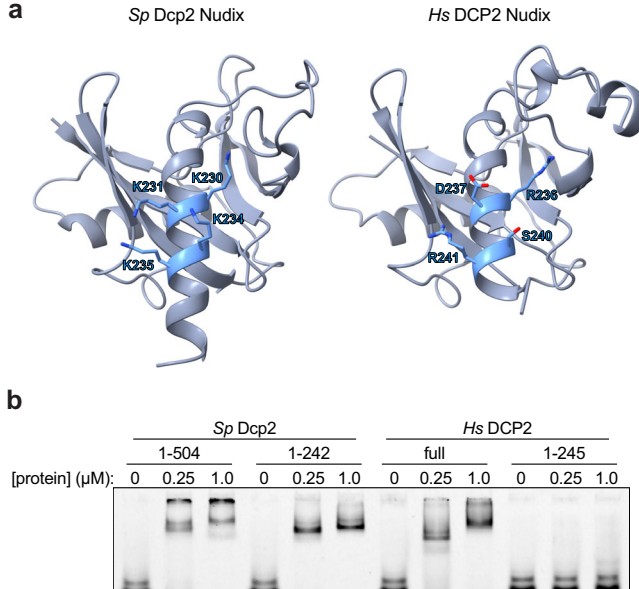

**b**

Fig. 3 | The C-terminal intrinsically disordered region boosts the RNA-binding capacity of DCP2. a Left, crystal structure of the *Sp* Dcp2 Nudix domain (PDB 2QKM) with the four lysines that mediate Box B RNA binding (Lys230, Lys231, Lys234, Lys235) shown as blue sticks. Right, AlphaFold model of the human DCP2 Nudix domain (AF-Q8IU60-F1) with the residues occupying the equivalent positions (Arg236, Asp237, Ser240, Arg241) rendered as sticks. Backbones are colored slate in both views. b Electrophoretic mobility-shift assays (EMSA) were performed with a 5′-labeled Rrp41 2 × DE RNA probe and increasing concentrations (0, 0.25, 1.0 μM) of the indicated DCP2 constructs. Full-length *Sp* Dcp2(1–504) and human DCP2 produce robust shifts, whereas their catalytic cores, lacking most or all of the C-terminal IDR, *Sp* Dcp2(1–242) and *Hs* DCP2(1–245), display markedly reduced binding. The gel shown is representative of three independent experiments.

involvement of C-terminal IDRs in substrate recognition, we tested the RNA-binding activity of the DCP2 constructs used in the initial decapping experiments using an electrophoretic mobility shift assay (EMSA) (substrate in Supplementary Table 1) (Fig. 3b). Removing the C-terminal region of yeast Dcp2 did not reduce RNA binding, indicating that the structured N-terminal domains are sufficient for substrate recognition. The shift in migration of bound RNA seen after removing the C-terminal IDR represents a reduced portion of the protein-RNA complex retained in the well. In contrast, truncating human DCP2 nearly eliminated RNA-binding activity, suggesting that the charge-enriched IDR plays a critical role in substrate recognition. The C-terminally truncated *Hs* DCP2 preserves the Nudix catalytic/regulatory core needed for transient 5′-cap engagement but lacks the positively charged C-terminal tail that confers high-affinity RNA-body binding. This may explain why this construct fails to form the stable complexes detected by EMSA yet remains capable of cap cleavage in decapping assays.

We next used the IDR sequence analysis tool flDPnn[39] to predict RNA-binding propensity along the length of the human DCP2 C-terminal IDR (Supplementary Fig. 1b). This analysis identified several segments of the DCP2 C-terminal IDR with high propensities for RNA binding. This finding marks a significant divergence in the basic mechanisms underlying decapping activity, suggesting the human C-terminal IDR has an emergent role in substrate recognition. To further explore how the human C-terminal region might facilitate substrate recognition, we used RoseTTAFold2NA[40] to generate structural predictions of full-length human DCP2 in complex with RNA substrate (Supplementary Fig. 1c–e). The highly charged IDR was consistently predicted to interact with RNA, which is sandwiched between the

charged IDR and a surface formed by both the Nudix and NRD domains (Supplementary Fig. 1f).

## Trimerization dynamics and interaction specificity of human DCP1

To further investigate the mechanisms of decapping regulation, we produced recombinant human DCP1 in its full-length form for the first time. Previous work reported trimerization of an isolated peptide sequence from the C-terminal region of DCP1, termed the trimerization domain (TD)[17] (Fig. 1a). To determine whether trimerization also occurs in the context of full-length DCP1, we used mass photometry, which directly estimates the molecular mass of individual protein particles in solution. DCP1 particles were primarily ~190 kDa, consistent with the theoretical mass of trimers (Fig. 4a, upper). To test whether the previously described TD was sufficient to mediate trimerization, we generated a DCP1 construct with mutations that prevent trimerization of the TD peptide[17]. Mass photometry measurements of the DCP1 TD mutant (TDm) indicated a primarily monomeric form (Fig. 4a, lower), confirming the importance of TD interactions in DCP1 trimerization. For this mutant, we also detected a minor but distinct population of particles matching the mass of DCP1 dimers (~130 kDa), suggesting homomeric interactions distinct from trimerization may occur. Size-exclusion chromatography revealed that wild-type DCP1 eluted as a single peak at 118–120 ml, indicative of an exclusive trimeric species. In contrast, the trimer-disrupting mutant (DCP1 TDm) produced an additional significant peak at 145 ml, consistent with a substantial fraction of the species shifting to a monomeric state (Supplementary Fig. 2a, b). Fractions across both peaks were analyzed by SDS-PAGE to verify the protein integrity. We examined earlier-eluting gel filtration fractions to test for the presence of additional oligomers for DCP1. Within an earlier-eluting wild-type DCP1 fraction, we found populations matching the molecular weights expected for trimers, hexamers, and nonamers (Fig. 4b), indicating that DCP1 trimers can further associate to form higher-order oligomers. An earlier-eluting DCP1-TDm fraction contained monomers, dimers, and a small population of trimers (Fig. 4c).

In contrast to the stable complex formed between yeast Dcp2 and Dcp1[16,25], immunoprecipitation (IP) experiments suggest the interaction between human DCP2 and DCP1 is relatively weak[13,18]. Structural characterization of the yeast Dcp2:Dcp1 heterodimer[16] revealed residues critical to the interface, which are not conserved in human DCP1 (Supplementary Fig. 3). To understand how DCP1 influences DCP2 in the human system, we directly compared the yeast and human interactions in the absence of enhancers. We used a streptavidin-based pulldown assay to immobilize yeast and human DCP2 constructs and tested their ability to capture purified yeast and human DCP1, respectively. As expected, yeast Dcp2 efficiently captured Dcp1, with both truncated (1–242) and extended (1–504) constructs capturing Dcp1 to a similar extent (Fig. 5a). Consistent with prior IP results and the lack of interface conservation, neither full-length nor truncated (1–242) human DCP2 captured DCP1 (Fig. 5b). Known interactor PNRC2[27,28] efficiently captured DCP1, confirming the integrity of purified DCP1 (Fig. 5b).

We further examined these interactions using mass photometry. In isolation, yeast Dcp2(1–504) existed primarily as a monomer, but a small population of homodimers was also detected (Fig. 5c, upper). Although yeast Dcp1 is below the detection limit when measured alone, combining yeast Dcp1 with Dcp2(1–504) led to an increase in mass consistent with heterodimer formation (Fig. 5c, lower). Masses of Dcp2(1–504) monomers and homodimers increased by masses consistent with one and two Dcp1 molecules, indicating that both monomeric and homodimeric forms of yeast Dcp2 are capable of binding Dcp1. In isolation, human DCP1 and DCP2 were in trimeric and monomeric forms, respectively (Fig. 5d, upper and middle). In contrast to the yeast proteins, no shift in mass was observed upon combining human DCP1 and DCP2 (Fig. 5d, lower), indicating that no heterocomplexes were formed. Consistent

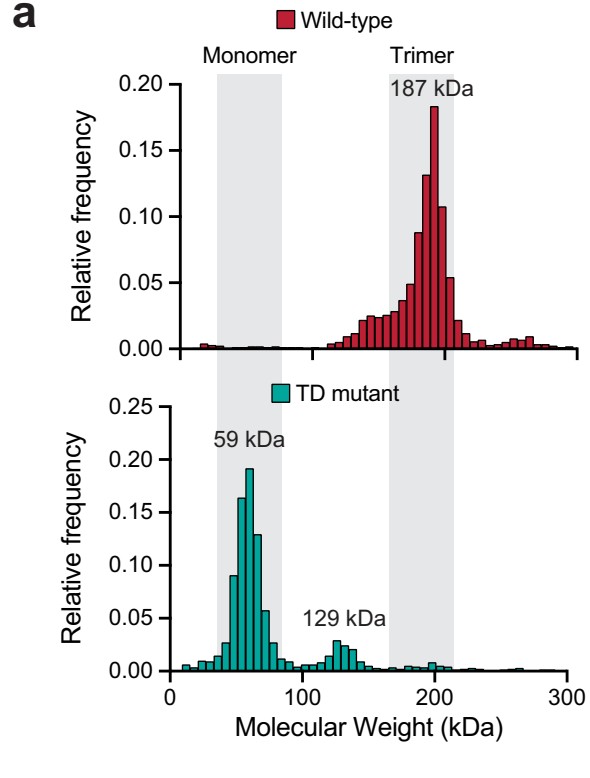

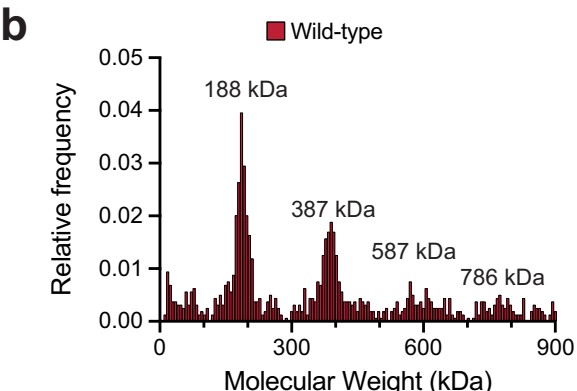

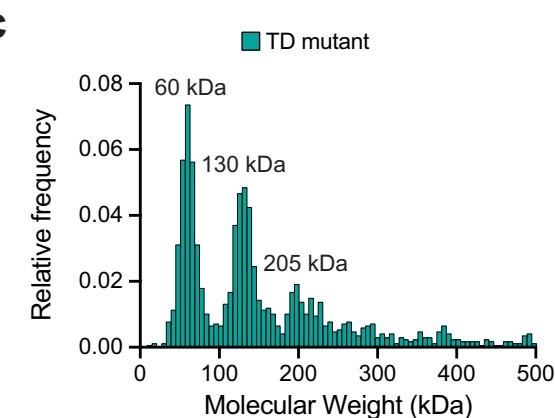

**Fig. 4 | The trimerization-domain (TD) mutant of human DCP1 abolishes stable trimer formation in solution. a** Mass photometry histograms show that wild-type *Hs* DCP1 (red) is almost exclusively trimeric, with a major peak at 187 kDa (gray band, "Trimer") and negligible monomer signal. In contrast, the TD mutant (teal) shifts to a predominant monomer peak at 59 kDa (gray band, "Monomer") plus a minor dimer shoulder at ~129 kDa. Representative replicate from $n = 3$ independent measurements. Peak labels denote estimates of apparent masses and may deviate modestly from exact integer multiples due to calibration/peak-picking. **b** Expanding the mass window reveals additional peaks at ~188 kDa (trimer), 387 kDa (hexamer), 587 kDa (nonamer), and 786 kDa (dodecamer), indicating stepwise self-association of trimers. Representative replicate, $n = 3$. **c** The mutant displays peaks at ~60 kDa (monomer), 130 kDa (dimer), and 205 kDa (trimer), but lacks species above 300 kDa, confirming that disruption of the TD suppresses higher-order assembly. Representative replicate, $n = 3$. In all panels, histograms are plotted as relative frequency versus calculated molecular weight, and shaded gray regions mark the expected mass windows for monomers and trimers.

impairing trimerization nor truncating DCP1 down to the EVH1 domain (1–135) altered the effect of DCP1 on DCP2 (Supplementary Fig. 4b, c). Similarly, adding yeast Dcp1 caused no increase in the activities of yeast Dcp2(1–242) or Dcp2(1–504) (Fig. 5f, Supplementary Fig. 4d, e). Additionally, activities of copurified yeast Dcp1/2 complexes were no higher than the basal activities of the corresponding Dcp2 constructs (Supplementary Fig. 4f).

To determine whether *Sp* Dcp1-mediated stimulation of *Sp* Dcp2 is metal-dependent, we removed $MnCl_2$ and retained $MgCl_2$ as the sole divalent cation in the reaction buffer. We next performed the assays in $Mg^{2+}$-only buffer to see whether the higher baseline activity with $Mn^{2+}$ had masked any Dcp1-specific stimulation. As expected, the overall decapping rate reduced in $Mg^{2+}$, prompting further investigation of whether the $Mn^{2+}$-enhanced turnover could be concealing a stimulatory effect of Dcp1 on Dcp2 (Supplementary Fig. 5a). We further lowered the Dcp2 concentration to 100 nM, quenched reactions at earlier time points, and supplemented Dcp1 at either equimolar or 10-fold molar excess. Although overall turnover decreased in $Mg^{2+}$, Dcp1 again produced no stimulation in equimolar ratio or excess (Supplementary Fig. 5b). Thus, any catalytic enhancement conferred by Dcp1 under these conditions appears to be minimal, and its absence is not an artifact of $Mn^{2+}$-dependent acceleration.

### DCP1 mediates stimulation of human DCP2 by PNRC2

Since DCP1 does not directly activate human DCP2, we explored whether it mediates DCP2 stimulation by decapping enhancers. We produced recombinant full-length PNRC2, which was proposed to function analogous to yeast enhancers Edc1 and Edc2[28], and assessed its effect on DCP2 activity with and without full-length DCP1 (Fig. 6a, Supplementary Fig. 6a). In the presence of DCP1, PNRC2 strongly stimulated decapping activity. However, adding PNRC2 in the absence of DCP1 decreased activity. Since PNRC2 directly interacts with RNA (Supplementary Fig. 6b), this reduction may result from RNA sequestration by PNRC2.

In yeast, the association of Edc1 with Dcp1 is necessary but not sufficient to stimulate the Dcp2:Dcp1 complex. The full decapping activating motif (DAM) of Edc1 includes a proline-rich sequence (PRS), which mediates the interaction with Dcp1, as well as a short segment N-terminal to the PRS. This segment contains a 'YAG' motif which binds the groove between the Dcp2 NRD and Nudix domains and is thought to stabilize an active conformation which positions the 5'-cap optimally in the active site[25,41,42]. Human PNRC2, as well as PNRC1, both contain a DAM that closely resembles the Edc1 DAM, including a YAG motif N-terminal to the PRS[43] (Fig. 6b).

To determine whether PNRC2 stimulates human DCP2 in a manner mirroring the yeast system, we compared the effect of PNRC2 PRS and DAM peptides to stimulation by full-length PNRC2 (Fig. 6c, Supplementary Fig. 6c). The effect of the PNRC2 DAM peptide matched that of full-length PNRC2, whereas stimulation by the PRS peptide

with our pulldown results, these data confirm that recombinant human DCP1 and DCP2 do not form a stable complex.

We next tested whether human DCP1 directly stimulates decapping activity by adding it to human DCP2 in decapping assays. We found that human DCP1 did not increase decapping beyond the basal activity of human DCP2 (Fig. 5e, Supplementary Fig. 4a). Neither

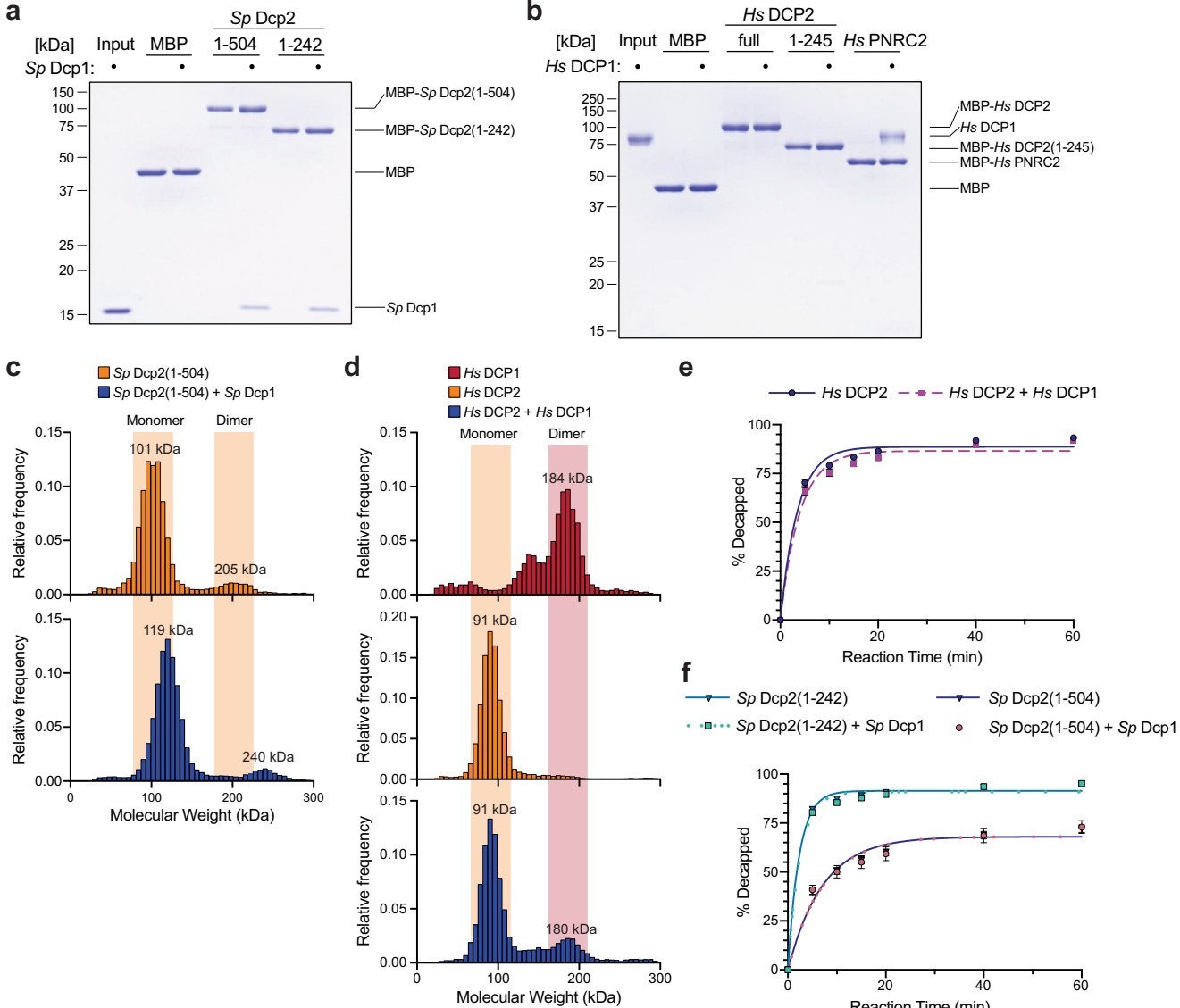

**Fig. 5 | DCP1 interaction with and effect on DCP2 in human and yeast homologs.**
**a** StrepII-tagged *Sp* Dcp2(1–504), *Sp* Dcp2(1–242), or maltose-binding protein (MBP) were immobilized on StrepTactin resin and incubated with purified *Sp* Dcp1. Bound material was eluted, resolved by SDS-PAGE, and visualized with Coomassie blue. The input lane shows *Sp* Dcp1 used as prey (representative gel from *n* = 3).
**b** Pulldowns were performed as in (**a**) using StrepII-tagged full-length *Hs* DCP2, the catalytic core *Hs* DCP2(1–245), MBP, or MBP-*Hs* PNRC2 as bait and purified *Hs* DCP1 as prey (representative gel, *n* = 3). **c** Mass photometry of *Sp* Dcp2(1–504) alone (top, orange) reveals a major monomer peak (101 kDa) and a minor dimer shoulder. Addition of equimolar *Sp* Dcp1 (bottom, navy) shifts the distribution to slightly higher masses (119 kDa, 240 kDa), consistent with formation of a heterodimer of one Dcp2 and one Dcp1. Shaded regions mark expected monomer and dimer windows (representative trace, *n* = 3). **d** Mass photometry histograms of *Hs* DCP1 alone (top, red) show a single trimeric species (~184 kDa). *Hs* DCP2 alone (middle,

orange) is monomeric (~91 kDa). A 1:1 mixture (bottom, navy) yields two populations: the DCP2 monomer and a species at ~180 kDa (pink shading) that matches the DCP1 trimer; no discrete DCP1–DCP2 complex is detected (representative trace, *n* = 3). Because each trimeric DCP1 particle contains three polypeptides, its particle count is lower than the equimolar DCP2 monomer. **e** Decapping assays were performed with full-length *Hs* DCP2 in the absence (blue) or presence (magenta) of equimolar *Hs* DCP1. Reactions were quenched at the indicated times (0–60 min), products were resolved by TLC, and the percentage of mRNA decapped was quantified. Data are mean ± SEM of three independent experiments (*n* = 3) with single-exponential fits. **f** Decapping kinetics of *Sp* Dcp2(1–242) (cyan) and *Sp* Dcp2(1–504) (light red) were measured with or without equimolar *Sp* Dcp1 (dotted lines). Plots show mean ± SEM of three independent experiments (*n* = 3) with single-exponential fits.

lacking the YAG motif was impaired. These findings demonstrate that DCP1 can mediate DCP2 stimulation by decapping enhancers and that the mechanism of stimulation by PNRC2 in humans mirrors that of Edc1 in yeast.

### The C-terminal region of EDC4 forms a tetrameric coiled-coil interface for DCP2

EDC4, a critical component of P-bodies, is thought to regulate the recruitment of DCP2 and other decay factors, such as the 5'-to-3'

exonuclease XRN1[30,33,44]. The C-terminal region of EDC4 contains an extended α-helical region necessary for self-association in cells[30,33]. The C-terminal 'distal' region of the *Drosophila melanogaster* EDC4 homolog has been structurally characterized[33], assuming a globular fold which consists of a bundle of α-helices and resembles ARM- and HEAT-repeat proteins. The 'proximal' region of EDC4, located N-terminal to the distal region, has not been structurally characterized.

To better understand how EDC4 influences mRNA decay, we recombinantly expressed and purified the C-terminal region (residues

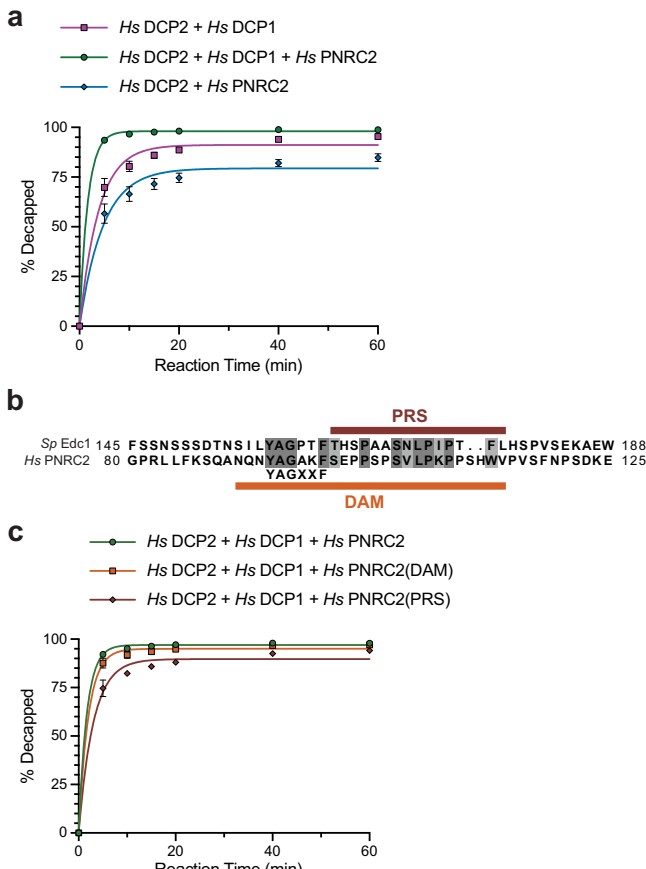

**Fig. 6 | PNRC2 stimulates human DCP2 through its Edc1-like decapping-activator motif. a** Decapping reactions were performed with *Hs* DCP2 and equimolar *Hs* DCP1, *Hs* PNRC2, or both cofactors. Aliquots were quenched at 0–60 min, products were resolved by TLC, and the percentage of substrate decapped was quantified. Plotted values are mean ± SEM of three independent experiments (*n* = 3); solid lines are single-exponential fits. **b** An alignment of the Edc1 activation segment (residues 145–188) and the corresponding PNRC2 region (residues 80–125) highlights the conserved YAGxxF decapping-activator motif (DAM, orange bar) and an adjacent proline-rich segment (PRS, maroon bar). Strictly conserved residues are shown in dark gray; functionally conserved are in light gray. **c** Decapping kinetics were measured for *Hs* DCP2 + DCP1 alone (baseline) and with either full-length *Hs* PNRC2, a synthetic DAM peptide, or a synthetic PRS peptide. Data were collected and fitted as in panel (**a**).

520–843) of the *Ce* (*Caenorhabditis elegans*) EDC4 homolog. Initial mass photometry measurements revealed that this C-terminal region exists primarily in a tetrameric state (Fig. 7a). Prompted by the observation that a small population consistent with the mass of octamers was also visible in initial measurements at 25 nM, increasing the concentration to 300 nM revealed the presence of higher order oligomers including 8-, 12-, 16-, 20-, 24-, and 28-mers (Fig. 7b). These results indicate that EDC4 exists in a base-level tetrameric state which is capable of assembling into larger oligomeric complexes. To explore the basis of tetramer formation, we used AlphaFold2-Multimer[45] to predict the structure of *C. elegans* EDC4 C-terminal region tetramers (Fig. 7c). Consistent with prior structural work, each distal portion was predicted to form a globular bundle of α-helices. The proximal regions, however, were predicted to assemble into an extended parallel tetrameric coiled-coil (Fig. 7d). Similar predictions for human and *Arabidopsis thaliana* homologs (Supplementary Fig. 7a, b) suggest that coiled-coil formation by C-terminal proximal regions could provide a conserved mechanism for EDC4 tetramer formation. The structure prediction of full-length *C. elegans* EDC4 revealed that WD40 domains are connected via flexible linkers to the coiled-coil region and are likely

to be dynamically flexible with respect to the C-terminal region (Supplementary Fig. 7c).

To experimentally validate this prediction, we crystallized a peptide from the proximal region of *Ce* EDC4 (residues 520–552). Molecular replacement revealed four α-helices arranged in a parallel tetrameric coiled-coil configuration, and the structure was refined to 1.4 Å resolution (Fig. 7e). The tetrameric assembly determined by crystallography superimposes well with the AlphaFold2-Multimer prediction (Fig. 7f, Supplementary Fig. 7d, e). To further characterize the architecture of the C-terminal region, we used negative stain electron microscopy to visualize the recombinant C-terminal construct. Micrographs showed extended particles with long filamentous regions flanked by globular domains (Fig. 7g). 2D classification revealed a central filamentous region ~25 nm long and an overall architecture consistent with the predicted tetrameric structure (Fig. 7h). Notably, numerous 'bent' particles in the micrographs suggest that the coiled-coil region is flexible.

Previous work has shown that DCP2 associates with the proximal C-terminal region of EDC4 via a phenylalanine-rich EDC4 binding (FEB) motif in the DCP2 C-terminal IDR[18]. To explore how EDC4 tetramers might interact with DCP2, we used AlphaFold2-Multimer to predict the structure of the EDC4 C-terminal region in complex with the C-terminal IDR of DCP2. Predictions for both *Hs* and *Ce* homologs showed the FEB motif of DCP2 binding along the surface of the EDC4 tetramer (Fig. 8a, b). These predictions locate key FEB motif phenylalanines within a cleft formed by EDC4 tetramerization. To test this interaction, we conducted a pulldown assay using the C-terminal region of *Ce* DCP2 (residues 745–786), predicted to mediate binding. This peptide successfully captured the C-terminal α-helical region of EDC4 (Fig. 8c), supporting the notion that the FEB motif of DCP2 interacts with the EDC4 tetrameric coiled-coil. Together, these findings suggest a conserved mechanism where the FEB motif binds into a groove assembled by the tetrameric proximal α-helical coiled-coil, explaining the critical role of FEB phenylalanines in the DCP2-EDC4 interaction.

It has been hypothesized that EDC4 enhances decapping by binding both DCP1 and DCP2, facilitating DCP1 stimulation of DCP2[18]. To test this, we purified recombinant full-length human EDC4 and conducted decapping assays. Contrary to the hypothesis, EDC4 did not stimulate DCP2 activity, whether DCP1 was present or not (Supplementary Fig. 8a, b). Instead, EDC4 addition to the decapping reaction consistently decreased activity. We then tested whether EDC4 can potentiate the DCP1/PNRC2-mediated stimulation of DCP2. Under conditions where PNRC2 robustly stimulates DCP2/DCP1, we observed that EDC4 does not provide additional catalytic enhancement (Supplementary Fig. 8c, d).

In conclusion, we identified significant functional differences between yeast and human DCP2, particularly regarding the roles of the C-terminal IDRs. The human C-terminal region is crucial for RNA substrate binding, whereas the yeast C-terminal region contains autoinhibitory motifs. Additionally, DCP1 does not directly stimulate DCP2 in either species but mediates stimulation by decapping enhancers. Furthermore, we demonstrated using the *Ce* ortholog that EDC4 forms a tetrameric structure that interacts with DCP2's C-terminal FEB motif, suggesting a conserved decapping regulatory mechanism among metazoans. These findings underscore the importance of species-specific adaptations in decapping regulation.

## Discussion

Decapping is a crucial regulatory step in 5'-to-3' mRNA decay, controlled by enhancers that modulate DCP2 activity. While much of our understanding of DCP2 regulation is derived from studies of the yeast system, research on human homologs has largely depended on immunopurified complexes and truncated recombinant proteins. In this study, we produced full-length recombinant human decapping factors, enabling direct in vitro examination of regulatory mechanisms

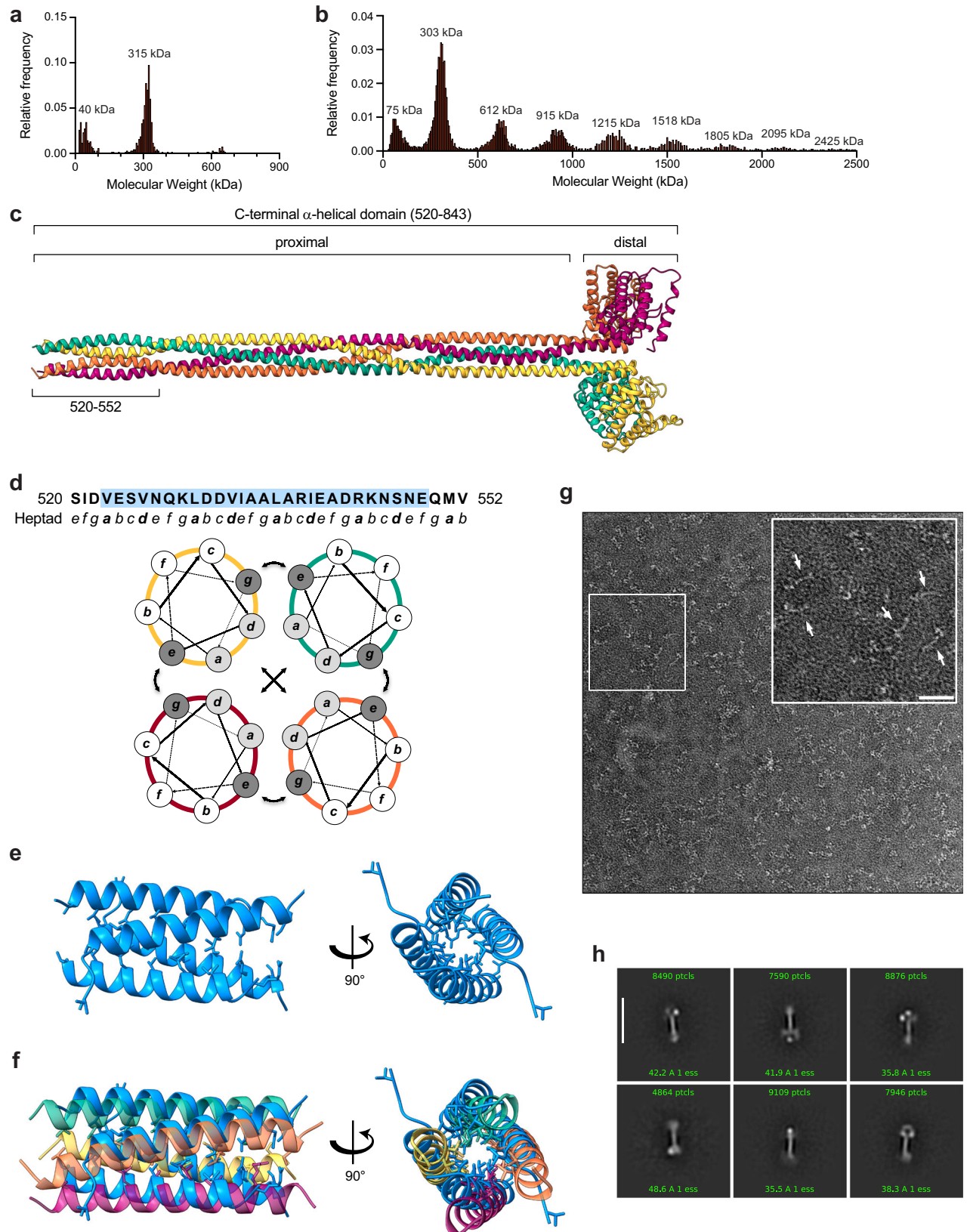

**d**

520 **SID**VESVNQKLDDVIAALARIEADRKNSNE**QMV** 552
Heptad *e f g **a** b c **d** e f g a b c **d**e f g **a** b c **d**e f g **a** b*

the human network utilizes a similar motif on DCP1 to recruit EDC3[46]. This redistribution reflects a reorganization of regulatory interactions in higher eukaryotes. Accordingly, other human decapping regulators may possess trans-acting motifs that inhibit DCP2 activity, compensating for the lack of autoinhibitory sequences in human DCP2.

Consistent with previous studies, we observe that the yeast Dcp2 C-terminal region exhibits autoinhibitory activity independently of Dcp1, whereas the human DCP2 C-terminal region does not. These findings suggest evolutionary shuffling of functional motifs within the decapping network. For example, while yeast Dcp2 contains an Edc3-binding motif,

Unlike yeast, we show that the C-terminal region of human DCP2 is essential for full basal decapping activity and RNA substrate interaction.

**Fig. 7 | Oligomeric state and structural organization of EDC4 C-terminal coiled-coil. a, b** Mass photometry histograms for purified *Ce* EDC4 residues 520–843 reveal a single species at ~315 kDa at 25 nM (**a**), consistent with a tetramer, and successive, equally spaced peaks beginning at ~303 kDa when the concentration is increased to 300 nM (**b**), indicating higher-order oligomers that grow in multiples of the tetrameric unit. Representative traces from three independent data sets are shown. **c** Residues 520–843 form an extended parallel four-helix bundle. The proximal stalk (rainbow-colored from green to yellow) is continuous with the distal α-helical bundle (magenta and cyan). The crystallized segment analyzed in panels (**d–f**) is bracketed (520–552). **d** Top, the 33-residue peptide (520–552) with residues resolved in the crystal structure colored blue; the canonical coiled-coil heptad positions (**a–g**) are annotated below. Bottom, helical-wheel diagrams illustrate the

packing of hydrophobic a/d residues (gray) within a parallel tetramer. **e** Two orthogonal views show the crystal structure of the segment of the parallel four-helix bundle; a/d layer residues are depicted as sticks. **f** Superposition of the crystal structure (blue) onto the corresponding region of the AlphaFold-Multimer model (rainbow) demonstrates near-perfect register and side-chain packing.
**g** Representative micrograph of negatively stained MBP-tagged *Ce* EDC4(520–843) with boxed inset (arrows mark individual particles); scale bar, 50 nm. The experiment was repeated independently three times with similar results. **h** Selected 2D class averages (particle counts and estimated resolutions indicated; scale bar, 51 nm) reveal elongated rod-shaped molecules terminating in a globular MBP density, consistent with a long coiled coil capped by the distal bundle.

While yeast Dcp2 uses BoxB residues for RNA binding[16,25], these residues are absent in human DCP2, and the charged C-terminal IDR appears to compensate. RNA recognition via disordered sequences has been previously observed[47], and this mode of interaction could contribute to the generality of DCP2 substrates. Further studies are needed to identify specific regions within the DCP2 IDR involved in RNA binding. Enhancer binding to motifs in this region may regulate substrate recognition, indicating a potential regulatory role. Unexpectedly, EDC4 reduced DCP2 activity, suggesting that EDC4 binding to the DCP2 IDR may interfere with substrate recognition. Consistent with this, EDC4 does not enhance catalysis even when PNRC2 is present, implying its primary role is likely architectural or regulatory rather than catalytic.

Previous studies of recombinant human DCP1 have primarily utilized C-terminally truncated constructs containing only the conserved EVH1 domain. In contrast, we produced full-length recombinant human DCP1 and discovered that it exists predominantly in a trimeric state. This trimerization is mediated by a specific motif known to form trimers in peptide form and to facilitate homomeric interactions in vivo[17]. Additionally, we observed the formation of higher-order multimers composed of DCP1 trimers and identified homomeric interactions independent of the trimerization motif, which may further contribute to oligomerization. This propensity for oligomerization could play a significant role in the function of DCP1 within P-bodies, as multivalency is a crucial factor in condensation processes, and yeast Dcp1 was recently shown to form condensates[48,49]. DCP1b, a paralog of DCP1 (alternatively referred to as DCP1a), was recently shown to link the decapping network to protein degradation and translation[50]. Given the functional non-redundancy of these paralogs, it will be interesting to explore whether the biochemistry of DCP1b mirrors our findings.

In yeast, Dcp1 and Dcp2 form a stable complex, but the interface residues essential for this interaction are poorly conserved in human DCP1. Our experiments confirmed that recombinant human DCP1 and DCP2 do not form a stable complex and showed that DCP1 does not directly stimulate DCP2 activity in vitro. In our reconstituted *S. pombe* system, Dcp1 alone also does not stimulate Dcp2 with magnesium as a co-factor, which contrasts with the potent activation mediated by Edc1/2. However, because we used MBP-tagged constructs for biochemical tractability, we cannot rule out that the MBP tag alters DCP2's responsiveness to DCP1. These findings align with immunoprecipitation studies indicating that additional enhancers are necessary to facilitate the in vivo association of DCP1 with DCP2[18].

Despite the lack of direct interaction, we confirmed that the role of DCP1 as a mediator of DCP2 stimulation by enhancers is conserved in humans. Specifically, we found that PNRC2 stimulates DCP2 activity in a DCP1-dependent manner, mirroring the stimulatory effect of Edc1 observed in yeast. This suggests that PNRC2, which is known to recruit the decapping machinery to NMD targets[27], could enhance targeted transcript degradation by stimulating decapping activity. Further analysis revealed that full-length PNRC2 directly interacts with RNA, whereas the DAM motif alone is sufficient to stimulate decapping activity. Although PNRC2's RNA-binding ability is not necessary for the stimulation of DCP2, it may play a role in other stages of mRNA

recruitment or decay processes. This indicates that PNRC2 and PNRC1 may facilitate decapping through multiple mechanisms, contributing to the regulation of mRNA turnover.

EDC4 is an essential component of P-bodies and the decapping network[13,30,31,35,44]. While it is known that EDC4 self-associates within cells, the specific nature of these interactions has remained unclear. Our biochemical experiments reveal that EDC4 forms tetrameric structures through coiled-coil interactions in the proximal C-terminal α-helical region. Additionally, these tetramers associate to create higher-order oligomers. We detected soluble multimers with megadalton masses, and given the limitations of our detection method, it is plausible that even larger assemblies are formed. These insights advance our understanding of EDC4's structural organization and its function within the decapping network.

The C-terminal α-helical region of EDC4 is essential for its localization to P-bodies[30,51], indicating that coiled-coil interactions play a central role in P-body formation. Electron microscopy analysis has characterized P-bodies as clusters of electron-dense fibrils[52]. Given the coiled-coil structure and extensive multimerization observed in our studies, these fibrils likely represent higher-order EDC4 assemblies. Coiled-coil-mediated higher-order oligomerization is a known mechanism for facilitating functions in membraneless organelles[53]. Condensate dynamics are determined by a complex interplay of biophysical properties and therefore can be highly tuned and context dependent[54]. The wide range of self-associating species we observed suggests EDC4 could play a role in fine-tuning condensate-cytoplasm exchange. These higher-order EDC4 oligomers could also provide a multivalent platform for recruiting other mRNA decay factors, thereby regulating decay by increasing local concentrations or affecting activity through phase separation.

Furthermore, conserved residues within the DCP2 C-terminal FEB motif are crucial for interacting with EDC4[18,35]. Our findings suggest that the aromatic residues of the FEB motif bind within an interface formed by EDC4 tetramerization. Each EDC4 coiled-coil tetramer can theoretically accommodate four FEB motifs, thereby enhancing the multivalency of the EDC4 scaffold and potentially facilitating more efficient recruitment of mRNA decay factors.

To conclude, our study reveals significant differences in decapping regulation between human and yeast systems, particularly concerning the roles of DCP2's C-terminal region and DCP1. Additionally, we characterized the tetrameric structure of EDC4 and its capacity for higher-order assembly, providing valuable insights into conserved regulatory mechanisms of mRNA decay across metazoan species. Collectively, these findings underscore the importance of species-specific adaptations in decapping regulation and open new avenues for investigating the control of mRNA decay.

## Methods
### Cloning
Plasmids used for protein expression are detailed in Supplementary Table 2. Fragments generated by PCR with Phusion polymerase (NEB) and/or restriction digest were assembled using standard isothermal

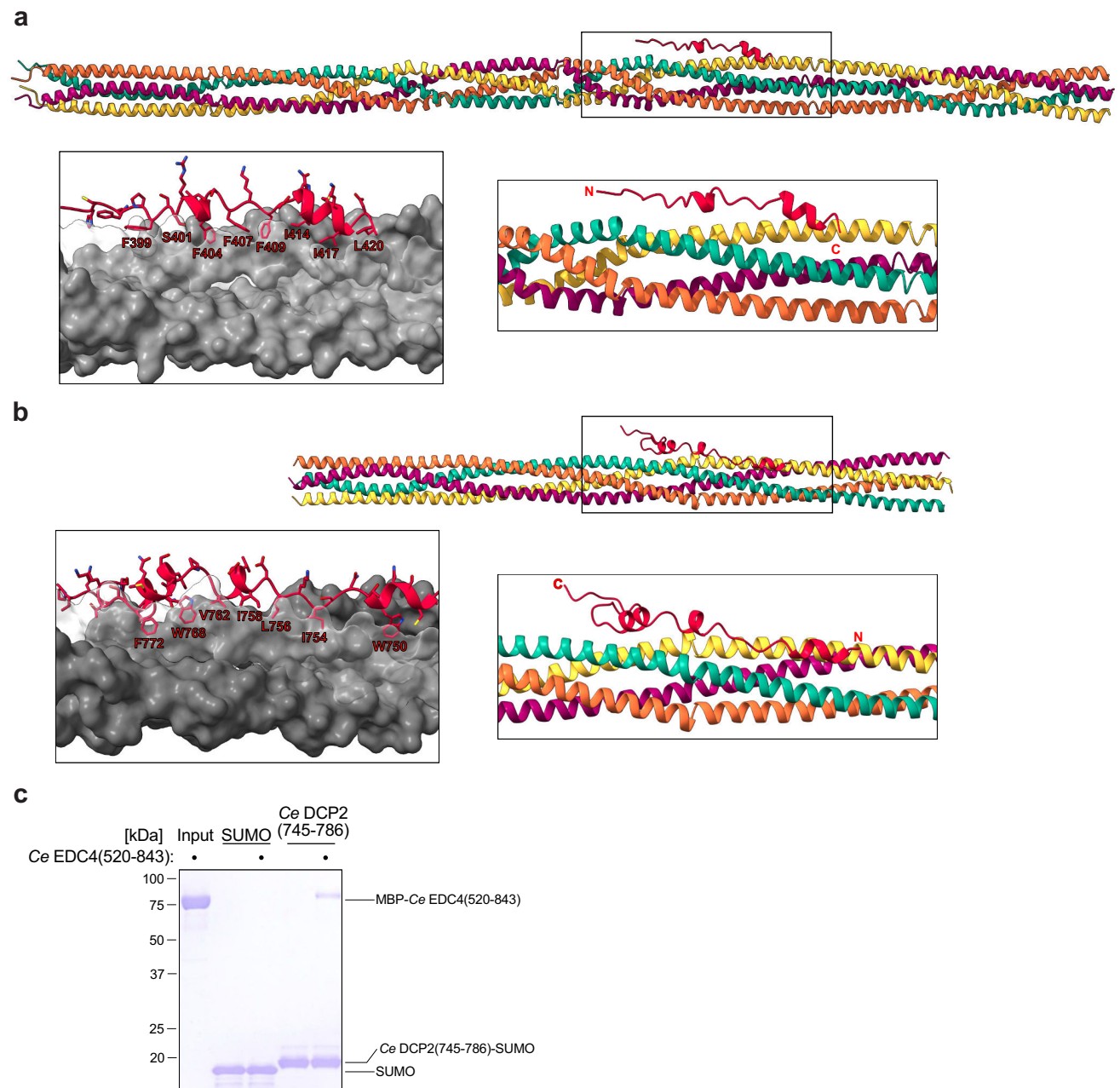

**Fig. 8 | A conserved phenylalanine-rich motif in the DCP2 C-terminus docks onto the tetrameric EDC4 coiled coil. a** An AlphaFold-Multimer was used to place the C-terminal IDR of *Hs* DCP2 (residues 393–420; red) onto the tetrameric coiled coil of the *Hs* EDC4 C-terminal α-helical region (residues 945–1250; four protomers colored cyan, yellow, magenta, and orange). The boxed region is enlarged (right) and rendered as a molecular surface (left inset) to highlight an extended phenylalanine-rich stretch in DCP2 (F399–L420) that packs against the hydrophobic groove formed by two adjacent EDC4 helices. **b** A corresponding AlphaFold-Multimer prediction shows the *Ce* DCP2 C-terminal fragment (residues 745–786; red) bound to the *Ce* EDC4 coiled coil (residues 520–697). The inset (left) reveals a similar array of bulky hydrophobics anchoring the interaction, consistent with the mechanism inferred from the human model. **c** StrepII-tagged *Ce* DCP2(745–786)-SUMO or a SUMO control was incubated with MBP-*Ce* EDC4(520–843). Pulldown eluates were analyzed by Coomassie-stained SDS-PAGE. MBP-*Ce* EDC4 co-purifies only with the DCP2 bait, confirming a direct interaction mediated by the predicted FEB segment (representative of *n* = 3).

assembly procedures, and assembled products were propagated in DH5α *Escherichia coli*.

## Baculovirus production
pLIB vectors described in Supplementary Table 2 were transformed into chemically competent DH10EmbacY cells (Geneva Biotech, Switzerland). Positive integrands were selected using blue/white screening, and bacmid DNA was purified as previously described[55]. 1–5 µg of bacmid DNA was transfected into $1.0 \times 10^6$ Sf21 cells (a kind gift from Imre Berger, University of Bristol) using Fugene HD (Promega) and

supernatant containing $V_0$ low-titer baculovirus was collected once at least 50% of cells were positive for YFP marker expression. Suspension cultures at $1.0 \times 10^6$ cells/mL were infected with 10% (v/v) $V_0$ baculovirus stock and high-titer $V_1$ baculovirus stocks were collected 24 h following proliferation arrest.

## Protein expression and purification
Full-length human DCP2 with an N-terminal MBP fusion and C-terminal His$_6$ tag was expressed in BL21 Star (DE3) *E. coli*, induced with 0.5 mM IPTG for 3 h at 30 °C. Cells were lysed by sonication in a buffer

containing 50 mM HEPES/NaOH pH 7.0, 300 mM NaCl, 2.5 mM CHAPS, and 5% (v/v) glycerol, and lysate was clarified by centrifugation. Affinity capture was performed with amylose resin (NEB) and then IMAC using a HisTrap HP column (Cytiva), followed by size exclusion chromatography on a Superdex 200 26/600 column (Cytiva) equilibrated in lysis buffer. Purified protein was concentrated with a 30 K MWCO centrifugal filter unit (MilliporeSigma), flash-frozen, and stored at −80 °C. For pulldowns, full-length human DCP2 with an N-terminal MBP fusion and C-terminal 2xStrepII tag was purified using the same procedure with the IMAC step omitted.

C-terminally-truncated human DCP2 (residues 1–245) with an N-terminal MBP fusion was expressed in BL21 Star (DE3) *E. coli* and induced with 0.5 mM IPTG for 3 h at 30 °C. Cells were lysed by sonication in a buffer containing 50 mM HEPES/NaOH pH 7.0, 300 mM NaCl, and 5% (v/v) glycerol, and lysate was clarified by centrifugation. Amylose resin (NEB) was used for affinity purification, followed by size exclusion chromatography on a Superdex 200 26/600 column (Cytiva) equilibrated in lysis buffer. Purified protein was concentrated with a 30 K MWCO centrifugal filter unit (MilliporeSigma), flash-frozen, and stored at −80 °C.

*Sp* Dcp2 including a portion of C-terminal IDR sequence (residues 1–504) with a 3C-cleavable N-terminal MBP fusion and C-terminal 2xStrepII tag was expressed in Sf21 cells. Cells were infected with recombinant *Sp* Dcp2(1–504) baculovirus stock (1:100 v/v) and harvested 48 h following proliferation arrest. Cells were lysed by sonication in a buffer containing 50 mM Tris/HCl pH 7.5, 300 mM NaCl, 2.5 mM CHAPS, 5% (v/v) glycerol, and 1 mM TCEP. Lysate was clarified by centrifugation and filtered through 0.45 μm syringe-driven filters (MilliporeSigma) before affinity purification with a StrepTrap HP column (Cytiva). Protein was eluted in lysis buffer supplemented with 2.5 mM desthiobiotin, concentrated with a 30 K MWCO centrifugal filter unit (MilliporeSigma), flash-frozen, and stored at −80 °C. C-terminally truncated *Sp* Dcp2 (residues 1–242) with an N-terminal MBP fusion and C-terminal 2xStrepII tag was expressed in BL21 Star (DE3) *E. coli*, induced with 1 mM IPTG for 16 h at 20 °C. *Sp* Dcp2(1–242) was purified using the same procedure as *Sp* Dcp2(1–504) with the concentration step omitted.

*Sp* Dcp1 with a C-terminal His$_6$ tag was expressed in BL21 Star (DE3) *E. coli*, induced with 0.5 mM IPTG for 3 h at 30 °C. Cells were lysed by sonication in a buffer containing 50 mM HEPES/NaOH pH 7.0, 500 mM NaCl, 2.5 mM CHAPS, 5% (v/v) glycerol, and 20 mM imidazole. Lysate was clarified by centrifugation and filtered through 0.45 μm syringe-driven filters (MilliporeSigma) before IMAC purification with a HisTrap HP column (Cytiva). Following elution, a HiPrep 26/10 desalting column (Cytiva) was used for buffer exchange into storage buffer containing 50 mM HEPES/NaOH, pH 7.0, 500 mM NaCl, 2.5 mM CHAPS, and 5% (v/v) glycerol before the protein was flash-frozen and stored at −80 °C.

Copurified complexes *Sp* Dcp1:*Sp* Dcp2(1–242) and *Sp* Dcp1:*Sp* Dcp2(1–504) were produced by co-expression of *Sp* Dcp1 with each *Sp* Dcp2 construct in BL21 Star (DE3) *E. coli*, induced with 0.5 mM IPTG for 16 h at 16 °C. Cells were lysed by sonication in a buffer containing 50 mM HEPES/NaOH, pH 7.5, 300 mM NaCl, 2.5 mM CHAPS, 5% (v/v) glycerol, 20 mM imidazole, and 1 mM TCEP. Lysate was clarified by centrifugation and filtered through 0.45 μm syringe-driven filters (MilliporeSigma) before IMAC purification with a HisTrap HP column (Cytiva). Complexes were further purified by size exclusion chromatography using a Superdex 200 26/600 column (Cytiva) equilibrated in storage buffer containing 50 mM HEPES/NaOH, pH 7.5, 300 mM NaCl, 2.5 mM CHAPS, 5% (v/v) glycerol, and 1 mM TCEP before protein was flash-frozen and stored at −80 °C.

Full-length *Hs* DCP1 (alternatively referred to as DCP1a) with an N-terminal His$_6$ tag was expressed in Sf21 cells. Cells were infected with recombinant DCP1 baculovirus stock (1:100 v/v) and harvested 48 h following proliferation arrest. Cells were lysed by sonication in a buffer containing 50 mM Tris/HCl, pH 8.0, 500 mM NaCl, 2.5 mM CHAPS, 5%

(v/v) glycerol, and 20 mM imidazole. Lysate was clarified by centrifugation and filtered through 0.45 μm syringe-driven filters (MilliporeSigma) before IMAC purification with a HisTrap HP column (Cytiva). Protein was further purified by size exclusion chromatography using a Superdex 200 26/600 column (Cytiva) equilibrated in storage buffer containing 50 mM HEPES/NaOH, pH 7.0, 300 mM NaCl, 5% (v/v) glycerol, and 2 mM TCEP before flash-freezing and storage at −80 °C. The DCP1-TDm mutant was expressed and purified using the same procedure as the wild-type construct.

C-terminally-truncated *Hs* DCP1 (residues 1–135) with a 3C protease-cleavable N-terminal MBP fusion was expressed in BL21 Star (DE3) *E. coli*, induced with 1 mM IPTG for 3 h at 30 °C. Cells were lysed by sonication in a buffer containing 50 mM HEPES/NaOH, pH 7.5, 500 mM NaCl, 2.5 mM CHAPS, 5% (v/v) glycerol, and 1 mM TCEP, and the lysate was clarified by centrifugation. Affinity purification was carried out using amylose resin (NEB) and followed by 3C protease cleavage, incubating with 1:30 (w/w) 3C at 4 °C for 16 h. Cleaved DCP1(1–135) was separated from MBP by size exclusion chromatography using a Superdex 200 26/600 column (Cytiva) equilibrated in lysis buffer. Purified protein was concentrated with a 10 K MWCO centrifugal filter unit (MilliporeSigma), flash-frozen, and stored at −80 °C.

Full-length *Hs* PNRC2 with an N-terminal MBP fusion was expressed in BL21 Star (DE3) *E. coli*, induced with 1 mM IPTG for 4 h at 30 °C. Cells were lysed by sonication in a buffer containing 50 mM HEPES/NaOH, pH 7.0, 500 mM NaCl, and 5% (v/v) glycerol, and the lysate was clarified by centrifugation. Affinity purification was carried out using amylose resin (NEB) and followed by size exclusion chromatography using a Superdex 200 26/600 column (Cytiva) equilibrated in lysis buffer. Purified protein was concentrated with a 30 K MWCO centrifugal filter unit (MilliporeSigma), flash-frozen, and stored at −80 °C.

Full-length *Hs* EDC4 with a C-terminal 2xStrepII tag was expressed in Sf21 cells. Cells were infected with recombinant EDC4 baculovirus stock (1:100 v/v) and harvested 72 h following proliferation arrest. Cells were lysed by sonication in a buffer containing 50 mM HEPES, pH 7.0, 300 mM NaCl, 2.5 mM CHAPS, and 5% (v/v) glycerol. Lysate was clarified by centrifugation and filtered through 0.45 μm syringe-driven filters (MilliporeSigma) before affinity purification with a StrepTrap HP column (Cytiva). Protein was eluted in lysis buffer supplemented with 2.5 mM desthiobiotin. Purified protein was flash-frozen, and stored at −80 °C.

N-terminally truncated *Ce* EDC4 (residues 520–843) with an N-terminal MBP fusion and C-terminal His$_6$ tag was expressed in BL21 Star (DE3) *E. coli*, induced with 0.5 mM IPTG for 16 h at 16 °C. Cells were lysed by sonication in a buffer containing 50 mM HEPES/NaOH, pH 7.0, 500 mM NaCl, 2.5 mM CHAPS, 10% (v/v) glycerol, 1 mM TCEP, and 25 mM imidazole. Lysate was clarified by centrifugation and filtered through 0.45 μm syringe-driven filters (MilliporeSigma) before IMAC purification with a HisTrap HP column (Cytiva). Following elution, a HiPrep 26/10 desalting column (Cytiva) was used for exchange into storage buffer containing 50 mM HEPES/NaOH, pH 7.0, 500 mM NaCl, 2.5 mM CHAPS, and 10% (v/v) glycerol before the protein was flash-frozen and stored at −80 °C. For electron microscopy, a construct with the His$_6$ tag replaced by a C-terminal 2xStrepII tag was expressed, lysed, and clarified in the same manner as above, but with imidazole omitted from the lysis buffer. Affinity purification was performed using a StrepTrap XT column (Cytiva), eluted in lysis buffer supplemented with 50 mM biotin, flash-frozen, and stored at −80 °C.

In all protein constructs that were produced as fusions with MBP, except for *Hs* DCP1(1–135), the N-terminal MBP tags were retained.

## Decapping assay
The RNA body was produced by in vitro transcription using T7 RNA polymerase (NEB) from a PCR-amplified DNA substrate using the manufacturer's protocol. The 144 nt product (Supplementary Table 1)

was purified by phenol:chloroform extraction followed by ethanol precipitation. Cap was added using Vaccinia Capping Enzyme and mRNA Cap 2′-O-Methyltransferase (NEB) with [α–$^{32}$P]GTP (PerkinElmer) following the manufacturer's 'one-step capping and 2′-O-methylation reaction' protocol to produce a cap-1 structure with $^{32}$P at the γ position of the cap's triphosphate linkage. Unincorporated nucleotides were removed using Sephadex G-50 spin columns (MilliporeSigma), and the capped substrate was further purified by phenol:chloroform extraction followed by LiCl precipitation. The capped RNA substrate was at ~40 nM (assuming 100% labeling efficiency) in all assays.

Proteins were pre-diluted and, in cases with multiple proteins, combined to 2 μM in 1X decapping buffer consisting of 50 mM Tris/HCl pH 7.5, 50 mM ammonium sulfate, 0.1% (w/v) BSA, and 5 mM MgCl$_2$ and incubated at 4 °C for 15 min. Reactions also contained 5 mM MnCl$_2$ (or MgCl$_2$ where indicated) and were initiated by adding proteins to a 200 nM final concentration (or as indicated). Reactions were carried out at 10 °C, and samples were taken at indicated time points and quenched by adding EDTA to 83 mM final concentration. Before sample application, PEI Cellulose F plates (MilliporeSigma) were pre-run in water and scored in 1 cm increments. 1 μl of quenched timepoint samples were applied, and plates were developed in 0.75 M LiCl. Minor differences in the insertion depth of each plate (±1–2 mm) altered the total solvent-front migration, resulting in a slight apparent offset in band position when the plates were scanned together.

TLC plates were exposed to BAS-IP MS storage phosphor screens (Cytiva) overnight, and screens were imaged using an Amersham Typhoon (Cytiva). Image quantification was performed in ImageJ (NIH), using the Gel Analyzer tool to plot signal intensity by lane. Authentic radiolabeled standards previously run in parallel confirm that the sole product comigrates with m$^7$GDP, not m$^7$GMP. This is also as expected for DCP2, which cleaves the β-γ phosphoanhydride bond to release m$^7$GDP. The product signal from $t = 0$ lanes was subtracted from the product signal of all lanes, and the percent decapped was calculated relative to the total signal from the product and intact substrate. Percents decapped were plotted by time point, and curves were produced in Prism 9 (GraphPad Software, Inc.) using nonlinear regression with least squares fitting using the 'One phase exponential association' equation.

### Electrophoretic mobility shift assay
RNA substrate for EMSA experiments consisted of a 60 nt sequence (Supplementary Table 1) from the 5′ UTR of the Rrp41 gene, which was previously found to interact with DCP2[56,57] and was synthesized commercially with a 3′ 6-FAM label (IDT). Proteins were pre-diluted in EMSA buffer consisting of 20 mM HEPES/NaOH, pH 7.5, 50 mM KCl, and 10% (v/v) glycerol. Binding reactions were conducted for 30 min at 4 °C with 10 nM RNA and 0.25–1.0 μM protein in EMSA buffer supplemented with 5 mM MgCl$_2$, 0.1 mg/mL BSA, and 0.1% (v/v) IGEPAL CA-630. 4% acrylamide (75:1 acrylamide:bisacrylamide) gels were cast and run in a buffer composed of 5 mM disodium borate decahydrate, pH 8.0 and 0.1% (v/v) IGEPAL CA-630. Binding reactions were loaded directly onto gels, run at 150 V for 5 min, and imaged using the Cy2 channel of an Amersham Typhoon (Cytiva).

### Pulldown assay
Bait proteins MBP-*Sp* Dcp2(1–242), MBP-*Hs* DCP2, MBP-*Hs* DCP2(1–245), MBP-*Hs* PNRC2, SUMO, and *Ce* DCP2(745–786)-SUMO fused to 2xStrepII tags were produced in BL21 Star (DE3) *E. coli* grown in auto-induction medium overnight at 37 °C. 2xStrepII-tagged MBP-*Sp* Dcp2(1–504) was expressed in the baculovirus system as described in the section 'Protein Expression and Purification'. Bacterial pellets from MBP-*Sp* Dcp2(1–242), MBP-*Hs* DCP2(1–245), and MBP-*Hs* PNRC2 expression were lysed by sonication in pulldown buffer containing

50 mM HEPES/NaOH, pH 7.5, 300 mM NaCl, and 2.5 mM CHAPS, and the lysate was clarified by centrifugation. Bait was loaded directly from lysate for all proteins other than 2xStrepII-tagged MBP and MBP-*Hs* DCP2, where purified proteins were diluted into pulldown buffer. 50 μl of StrepTactin Sepharose resin slurry was added to lysates or diluted bait solutions, and incubated for 1 h at 4 °C before washing three times with pulldown buffer. Purified *Sp* Dcp1 (40 μg) or *Hs* DCP1 (80 μg) was diluted in pulldown buffer and incubated with bait-loaded beads for 1 h at 4 °C before washing three times with pulldown buffer. Proteins were eluted with 50 μl of pulldown buffer supplemented with 50 mM biotin for 1 h at 4 °C and resolved by SDS-PAGE. Assays examining the interaction between *Ce* DCP2 and *Ce* EDC4 were performed as described above with a pulldown buffer containing 50 mM HEPES/NaOH, pH 7.0, 300 mM NaCl, 1 mM TCEP, and 0.03% (v/v) Tween 20, applying 80 μg of MBP-*Ce* EDC4(521–843) prey.

### Mass photometry
Measurements were acquired with the TwoMP mass photometer (Refeyn), using Refeyn AcquireMP software and calibrated as per the manufacturer's protocol using β-amylase and thyroglobulin. For measurements of DCP1 oligomerization, proteins were pre-diluted into and measured in a buffer containing 50 mM HEPES/NaOH, pH 7.0, 300 mM NaCl, 5% (v/v) glycerol, and 2 mM TCEP. To examine trimerization of wild-type and TDm mutant DCP1, purified protein stocks were diluted to 100 nM immediately before use, and 5 μl of 100 nM stock was added to 15 μl buffer droplets for measurement. To examine high molecular weight species isolated during size exclusion chromatography, 0.5 μl of void peak samples were diluted directly into 18 μl buffer droplets for measurement. For measurements of DCP2–DCP1 complexes, the buffer contained 50 mM HEPES/NaOH, pH 7.0 and 300 mM NaCl. Binding reactions were prepared in this buffer with proteins at 2 μM and incubated for 30 min at 4 °C. 1 μl of binding reactions were diluted directly into 19 μl buffer droplets for measurement. MBP-*Ce* EDC4(520–843)-His$_6$ was diluted into and measured in a buffer containing 50 mM HEPES/NaOH, pH 7.0 and 500 mM NaCl. For measurements at 25 nM, 5 μl of freshly diluted 100 nM protein solution was diluted into 15 μl droplets. To measure at 300 nM, 6 μl of 1 μM protein solution was diluted into 14 μl droplets. Frequency distributions were created from event counts in Prism 9 (GraphPad Software, Inc.), tabulating relative frequency with a bin width of 6. Molecular masses shown above peaks in frequency distribution plots were generated through analysis with Refeyn DiscoverMP software.

### Structural predictions
The AlphaFold prediction of human DCP2 (AF-Q8IU60-F1) was obtained from the AlphaFold Protein Structure Database and was created with the AlphaFold Monomer v2.0 pipeline[58,59]. Structural predictions of protein complexes were generated using AlphaFold2-Multimer, and each prediction shown is representative of five or more replicates[45]. Predictions of DCP2–RNA substrate complexes were generated using RoseTTAFold2NA[40]. We modeled the RNA substrate using a fragment of Me31B RNA, the substrate used in decapping assays, and Rrp41 2xDE RNA, the substrate used in gel shift assays (sequence provided in Supplementary Table 1), as well as a randomly generated 40-nt RNA sequence. The *Ce* EDC4 crystal structure alignment with the AlphaFold Multimer prediction was performed using the cealign command in PyMOL (The PyMOL Molecular Graphics System, Version 3.0, Schrödinger, LLC.). Structural images were prepared with UCSF ChimeraX[60].

### Bioinformatic analysis
Protein sequences were obtained from UniProt. The EMBOSS v6.5.7 tool 'charge' was used to calculate charge values with a window size of one. Protein disorder was predicted using IUPred2A[61], and the RNA-binding propensity of disordered regions was predicted with flDPnn[39].

Plots of amino acid charge and disorder score were generated with second-order smoothing and a sliding window size of 30. All plots were generated in Prism 9 (GraphPad Software, Inc.).

## Crystallization and structure determination

Crystals of *Ce* EDC4 (UniProt ID: Q8ITV7) C-terminal proximal region residues 520–552 were obtained at 18 °C using the sitting-drop vapor diffusion method two days after mixing 10 mg/mL peptide solution with crystallization solution containing 3 M ammonium sulfate and 10% glycerol. Crystals were cryoprotected in mother liquor supplemented with 4 M sodium formate and flash-cooled in liquid nitrogen. Data was collected at 100 K on a PILATUS 6 M detector at the PII beamline at the PETRAIII synchrotron in Hamburg, Germany. Diffraction data were processed with XDS and scaled using XSCALE[62]. Phases were obtained by molecular replacement using PHASER[63]. An idealized polyalanine helix was used as a search model with an asymmetric unit containing two copies of the model. The molecular replacement solution was used to rebuild the initial model using the PHENIX AutoBuild wizard[64]. To complete the structure, iterative cycles of model building and refinement were performed with COOT[65] and PHENIX[66], respectively. Stereochemical properties were verified with MOLPROBITY[67], and structural images were prepared with UCSF ChimeraX[60]. Diffraction data and refinement statistics are summarized in Supplementary Table 3.

## Negative-stain electron microscopy

MBP-*Ce* EDC4(520–843)–2xStrepII was diluted to 25 μg/mL in a buffer containing 50 mM HEPES pH 7.0 and 500 mM NaCl, and 5 μl was applied to freshly glow-discharged 400-mesh copper grids with carbon support (Electron Microscopy Sciences) and incubated at room temperature for 30 s. Grids were rinsed briefly twice, first with water and then 2% (w/v) uranyl acetate (UA), stained with two rounds of 2% (w/v) UA, first for 30 s and then for 1 min, then blotted and air-dried for at least 30 min before imaging. Images were acquired on a Talos L120C TEM with a 4k x 4k Ceta CMOS camera (Thermo Fisher Scientific). Data were acquired using SerialEM at a defocus of −5 μm at 57,000x magnification, corresponding to a pixel size of 4.83 Å/pixel.

All image processing was performed using cryoSPARC on a dataset of 98 micrographs. Blob Picker was used to pick circular and elliptical blobs with a 30–300 Å diameter. Micrographs were extracted with a 256-pixel box size, 148,073 particles were used for 2D classification and poor particles were removed, yielding 30,333 particles in 11 classes. Of these, six representative classes are shown.

## Reporting summary

Further information on research design is available in the Nature Portfolio Reporting Summary linked to this article.

## Data availability

Atomic coordinates and structure factors for the reported crystal structure have been deposited with the Protein Data Bank under accession code 9E7Q. Source data for the figures and Supplementary Figs are provided as a Source Data file. The corresponding authors will provide the raw data, additional information, and materials upon request and subject to a completed Materials Transfer Agreement. Source data are provided with this paper.

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

## Acknowledgments

Initial experiments were done in Elisa Izaurralde's former laboratory at the Max Planck Institute for Developmental Biology, Tübingen, Germany, and we gratefully acknowledge Elisa's support and encouragement during the project's initial phase. We acknowledge DESY (Hamburg, Germany) for the provision of experimental facilities at beamline P11, PETRA III. We also thank Sergey Tarasov and Marzena Dyba for support with biophysical measurements, Dan Shi for advice and guidance on electron microscopy, and Catrin Weiler for excellent technical assistance with cloning and insect cell culture. We are also grateful to our colleagues at the RNA Biology Laboratory for support and advice. This study was supported by the Intramural Research Program of the National Institutes of Health (project number 1ZIABC011977 to E.V.) and the Max Planck Society (E.V. and S.M.). The contributions of the NIH authors are considered works of the United States Government. The findings and conclusions presented in this paper are those of the authors and do not necessarily reflect the views of the NIH or the U.S. Department of Health and Human Services.

## Author contributions

E.A.J.S., S.M., T.M.M., and A.L.V. performed the experiments, S.M., E.A.J.S., and E.V. designed the project, E.V. supervised it, and E.A.J.S. wrote the initial draft. All authors contributed to manuscript preparation.

## Competing interests

The authors declare no competing interests.
