## [Transparent Peer Review file · Nature Communications]

Conserved and Divergent Features of Human mRNA Decapping Revealed by Biochemical Reconstitution

Corresponding Author: Dr Eugene Valkov

Version 1:

Reviewer comments:

Reviewer #1

(Remarks to the Author)

The manuscript by Simko et al uses biochemical reconstitution and structural approaches to determine the mechanisms of how protein interactions influence the enzymatic activity of the human decapping complex. While much prior genetic, biochemical, and structural data have been acquired on yeast decapping complex, the manuscript by Simko et al shows that the human complex has some similarities and striking differences, indicative of more complex regulatory mechanisms.

They convincingly show that the C-terminal IDR of human Dcp2 accelerates decapping, in contrast to the C-terminal IDR of fission yeast Dcp2 which is autoinhibitory. EMSAs reveal that the C-terminus of hDcp2 mediates binding to RNA whereas RNA binding is localized to the structure cores domains of fungal Dcp1/Dcp2 as previously described.

Using Mass Photometry (MP), they show that human Dcp1 is a trimer and exists as higher order species (a dimer of trimers). hDcp1 is not able to associate with hDcp2, assayed by pull-down and MP assays in contrast to yeast Dcp1/Dcp2 which forms a stable complex.

Simko et al show one role of hDcp1 is bind PNRC2 which stimulates decapping of Dcp1/Dcp2 *in vitro*. They show conserved motifs in PNRC2 shared by yeast enhancer of decapping proteins (Edc1 and Edc2) are responsible for this effect. Thus, a conserved role of Dcp1 is to recruit 'coactivators' of decapping such as PNRC2 to Dcp2.

Finally, the authors present structural and biochemical data on Edc4, a enhancer of decapping found in metazoans. They find, contrary to expectation that Edc4 inhibits decapping. Furthermore, they show it exists as a tetramer in solution and provide negative-stain EM and crystallographic analyses to support predictions made by AlphaFold.

Overall this manuscript is clearly-written and the data are of high-quality. Multiple approaches are used to corroborate the central findings (e.g. MP and PD assays, ML and experimental structure prediction). The manuscript is suitable for publication in Nature Communications after the following points are addressed.

1-On line 263 the authors state "Dcp1 does not directly stimulate Dcp2 in either species but mediates stimulation by decapping enhancers." This statement is not entirely supported by their data and the published literature. Yeast Dcp1 was shown to stimulate Dcp2 by around 10-fold (PMID:22323607). I doubt this stimulation could be observed in the kinetic analyses reported in Figure 5f due to the paucity of data in the initial phase of the exponential. To make this claim, the authors should accurately quantify decapping rates by recording a handful of timepoints in the early phase of the exponential (between 0 and 5 minutes). In addition, the authors should take care that the same decapping buffer is employed with prior studies if comparisons are to be made. In particular, Mg²⁺ is sufficient for decapping by yeast Dcp1/Dcp2; the addition of Mn²⁺ to the buffer will accelerate the rate of the chemical step such that the fraction of the closed form of the enzyme is no longer rate-limiting, potentially masking the effect of Dcp1. Alternatively, the authors should tone down their claim that yeast Dcp1 does not affect decapping by Dcp2.

2-The authors should clarify which experiments were performed without MBP tags in the Methods section. I could only find the details of tag removal for hDcp1.

3-The authors may wish to mention the tumor suppressor PNR1 has a DAM and probably acts similarly to PNR2 (PMID: 30373810).

4-Edc4 has a WD40 domain. Why was this omitted from the alpha-fold structure prediction?

Reviewer #2

(Remarks to the Author)

Much of the field's understanding of mRNA decapping is derived from studies of the yeast Dcp2 decapping protein and its activity/stimulation. Relatively less is known in metazoan systems. This manuscript uses a reconstituted system to characterize the decapping by homo sapien DCP2 decapping protein and stimulation of its activity. Importantly, they show that unlike yeast Dcp2, the decapping stimulator DCP1 does not directly interact with or stimulate DCP2 in humans. Interestingly, they show the interaction and stimulation is mediated by a third protein, PNR2 and further characterize the stoichiometry of the protein complex. The difference between the metazoan and yeast systems is an important addition to the field. The authors also study the metazoan restricted scaffold protein, enhancer of decapping 4 (EDC4) and its structure and activity. Contrary to previous reports, they show that the EDC4 does not stimulate DCP2 decapping. The data are sound and the findings interesting. Several points are detailed below to enhance the manuscript.

1. EDC4 has long been thought to serve as a scaffold that juxtaposes decapping factors to stimulate decapping. In Sup Fig 6, the authors show EDC4 does not stimulate DCP2 decapping and surprisingly inhibits it. They further show the EDC4 also inhibits decapping when DCP1 was included in the reaction. However, considering their findings that DCP1 does not interact with DCP2 and requires PNR2 to stimulate Dcp2 decapping, this later finding is not surprising. These studies should also be done with the inclusion of PNR2 to test whether EDC4 can further enhance the stimulation of DCP1/PNR2 on DCP2 decapping.

2. Markers on the TLC of Fig 2 should be included. Are the products m7GMP or m7GDP? Why does the product size shift over time?

3. Do the author have an explanation for why the DCP2 1-245 fragment does not bind RNA (fig 3b), but can still decap RNA (fig 2c)?

4. The identity of the various Short Functional Domains shown as black bars in Fig 1 should be defined. The phenylalanine-rich EDC4 binding (FEB) motif should also be defined.

Reviewer #3

(Remarks to the Author)

The publication by Simko & Muthukumar, Valcov describes a noteworthy effort to reconstruct human decapping mechanism in vitro using full length recombinant proteins. For many years the decapping reaction was extensively studied in vitro and in vivo using the budding and fission yeast models. Most key actors in this process are conserved or have identified counterparts in human. However, the considerable difference in amino acid sequence and the presence of additional domains in human proteins, suggests that the mechanistic details of human decapping might be very different from the yeast system. Therefore the work by Simko & Muthukumar, Valcov is a much needed step towards our full understanding of the decapping process.

The Introduction is well structured, supplemented with schemes, and clearly puts forward the known functions and mechanism of action of yeast decapping enzyme and its co-factors altogether putting forward the limited differences readily identifiable from evolutionary amino-acid sequence changes. Using a mix of approaches including in vitro assays using human recombinant proteins, protein interaction modeling and structural or semi-structural studies the Result section systematically characterizes the decapping reaction in vitro. The Discussion clearly sums up the in vitro results, that show a rearrangement of molecular functions in the human system compared to the yeast model. It also puts the multimerization data in the in vivo context of in vivo P-body formation; an aspect which is a bit more far fetched, but still acceptable in this section.

Altogether, the publication is engagingly written, filled with interesting data, and will be a good addition to Nature Communication portfolio. Given the rich collection of mechanistic data gathered in yeast, a systematic comparison to the human system is much needed in the decapping context, and beyond. I would therefore recommend the work for publication after the authors answer the few questions below.

Major:

Question 1:

Fig S1 - please present more clearly the hsDcp2 amino-acids presumably involved in RNA binding (Fig. S1f - number of aa type of aa), and if possible attempt to validate them by mutational analysis. What types of RNA sequences were used for

modeling? Are those 5'UTRs of actual mRNAs? Was a cap included?

Question 2:

Section: ,Trimerization Dynamics and Interaction Specificity of Human DCP1' - when preparing recombinant proteins for in vitro decapping experiments the authors performed size exclusion chromatography (as stated in M&M section). Was the multimerization visible using this technique? An orthogonal approach is always welcome in addition to mass photometry. Recombinant proteins like to create aggregates, thus I personally would downplay the conclusion about any higher (than a trimer) order structures until a clear motif responsible for their formation can be isolated. Even if this is highly interesting in the context of Dcp participation in P-bodies.

Best wishes in the New Year!

Version 2:

Reviewer comments:

Reviewer #1

(Remarks to the Author)

The revised manuscript by Simko addresses all of my questions except the one about the inconsistencies between the stimulatory effects of Dcp1 on Dcp2's catalytic activity reported in previous studies and their manuscript.

I appreciate their testing of the role of metal ions in decapping reactions; however, upon initial review, it was unclear whether decapping experiments were performed with MBP tags attached to Dcp2. The authors clarified in the revised manuscript that MBP tags were retained on the tested Dcp2 constructs. Therefore, the statement "Our findings reveal a divergence in the function of the DCP2 C-terminal sequence between yeast and human homologs, with no direct stimulation of DCP2 by DCP1 observed" on lines 86-87 is not supported by well-controlled experiments, as they have not ruled out the possibility that the latter result is an artifact of using MBP-tagged Dcp2. Indeed, prior studies showing a stimulatory effect of spDcp1 were performed with untagged spDcp2 (PMID:20711189; 22323607), and this conclusion is supported by a study from Valkov et al., which showed spDcp1 could stimulate RNA binding and decapping by spDcp2 (PMID:27183195, Supplemental Figure 4e, 4h). Notably, the Dcp1 and Dcp2 constructs used in the decapping assays reported by Valkov et al did not contain MBP or any other protein tags (PMID:27183195, Supplemental Figure 4i).

Thus, it seems likely that a source of discrepancy regarding Dcp1 stimulation of decapping by Dcp2 could be the retention of the MBP tag on Dcp2. This possibility should be investigated using EMSA and/or decapping assays with constructs that lack MBP tags. While Dcp1 can bind MBP-tagged Dcp2, this does not rule out the possibility that the MBP tag might interfere with RNA binding or catalytic enhancement, especially considering that this system undergoes conformational changes during catalysis (PMID:28533364).

Overall, this manuscript presents important new insights into the human decapping complex. I agree with the authors' assertion that the conserved function of Dcp1 is to recruit Edc1-type coactivators to stimulate Dcp2 catalysis. However, their claim that Dcp1 does not stimulate decapping in the absence of Edc1 or PNRC2 is based on experiments with MBP-tagged Dcp2 constructs, which could confound interpretation. The manuscript is worthy of publication, but the discrepancy with prior studies should be addressed either by additional experiments or by qualifying their conclusion about Dcp1's role to state 'the effect of the MBP tag on Dcp2's ability to be stimulated by Dcp1 cannot be ruled out' in the Discussion.

Minor:

1. The authors' explanation in the rebuttal that the inconsistency is due to 'lab to lab variation' is not correct. They mention, "Notably, earlier kinetic analyses were performed at considerably higher protein concentrations (1-10 μM SpDcp2/SpDcp1) than those used here (0.1 μM), a difference that likely contributes to lab-to-lab variability in the magnitude of Dcp1 stimulation." The effect of Dcp1 should be evident in the observed rate (kobs) measured under single-turnover conditions, where kobs is proportional to $k_{\text{max}}/K_{\text{m}}$ or k_{max} depending on whether the decapping enzyme concentration is below or above K_{m} , at low (100 nM) or high (10 μM) enzyme concentrations.

2. There is no section in the Methods describing how spDcp2 (1-242) was expressed and purified.

Reviewer #2

(Remarks to the Author)

The authors have adequately addressed my initial concerns, and I would recommend publication.

Reviewer #3

(Remarks to the Author)

The revised work by Simo et al. is a purely in vitro study that brings forward the evolutionary reshuffling of functions in the decapping machinery protein domains (DCP1/2, EDC4, PNRC2), mainly between the human and *S. pombe* systems. Additionally, the work provides experimental structural information about the EDC4 C-terminal coiled-coil domain, which mediates the protein's multimerization. Overall, the study gives much in vitro information to inspire follow-up in vivo studies. This I hope will be a much desired outcome of this work.

The authors have answered my previous questions and I think the publication should be accepted, with minor corrections to figures listed below. Those can be added in proof.

Minor:

Some of the introductory comparative experiments were performed with large fragments of *S. pombe* Dcp2, in contrast to the use of full-length human protein. The *S. pombe* protein is very large, which was likely the reason for its fragmentation. The decision and the origin of the construct should be better explained in the first paragraph of the results.

Fig. 1 - add numbers indicating length of depicted proteins and position of highlighted domains.

Fig. 4 - is the trimer 187 or 188 kDa? there is a discrepancy between fig. 4a and 4b? A 59 kDa monomer adds up to a 177 trimer. This whole panel can be a bit confusing. I understand that the method gives a mol. weight range. Maybe indicate the range of monomer and trimer size estimates in a smaller and grey font so that it becomes less confusing? The peak estimate can be left as it is.

Fig. 5d - the identity of the regions highlighted should be indicated as in fig. 5c

Fig. S1A - scale missing on x-axis (number of amino acids). This will enable to better compare panel a with panel b

Fig. S2 - please include a molecular weight scale for the SEC analysis (can be from the column's manual) in parallel to the elution volume. This will help to verify if the monomer size corresponds to the Mol. weight of DCP1.

AUTHOR RESPONSE TO REVIEWERS

We are grateful to the reviewers for their constructive and incisive comments. Their feedback substantially strengthened the manuscript and prompted several new lines of experimentation and analysis. Although our ability to conduct experiments was significantly hindered by issues such as prolonged delays in purchasing reagents and other unanticipated restrictions, we have repeated key assays, added orthogonal biophysical measurements, and expanded structural modeling as requested.

*All textual changes are marked in **dark red** in the main document, and our point-by-point replies follow in a **blue font** for ease of navigation.*

We trust that the revisions fully satisfy the reviewers' concerns and present a clearer, more comprehensive picture of the mechanisms of metazoan mRNA decapping and how these compare to observations in yeast.

Reviewer #1 (Remarks to the Author):

The manuscript by Simko et al uses biochemical reconstitution and structural approaches to determine the mechanisms of how protein interactions influence the enzymatic activity of the human decapping complex. While much prior genetic, biochemical, and structural data have been acquired on yeast decapping complex, the manuscript by Simko et al shows that the human complex has some similarities and striking differences, indicative of more complex regulatory mechanisms.

They convincingly show that the C-terminal IDR of human Dcp2 accelerates decapping, in contrast to the C-terminal IDR of fission yeast Dcp2 which is autoinhibitory. EMSAs reveal that the C-terminus of hDcp2 mediates binding to RNA whereas RNA binding is localized to the structure cores domains of fungal Dcp1/Dcp2 as previously described.

Using Mass Photometry (MP), they show that human Dcp1 is a trimer and exists as higher order species (a dimer of trimers). hDcp1 is not able to associate with hDcp2, assayed by pull-down and MP assays in contrast to yeast Dcp1/Dcp2 which forms a stable complex.

Simko et al show one role of hDcp1 is bind PNRC2 which stimulates decapping of Dcp1/Dcp2 in vitro. They show conserved motifs in PNRC2 shared by yeast enhancer of decapping proteins (Edc1 and Edc2) are responsible for this effect. Thus, a conserved role of Dcp1 is to recruit 'coactivators' of decapping such as PNRC2 to Dcp2.

Finally, the authors present structural and biochemical data on Edc4, an enhancer of decapping found in metazoans. They find, contrary to expectation that Edc4 inhibits decapping. Furthermore, they show it exists as a tetramer in solution and provide negative-stain EM and crystallographic analyses to support predictions made by AlphaFold.

Overall this manuscript is clearly-written and the data are of high-quality. Multiple approaches are used to corroborate the central findings (e.g. MP and PD assays, ML and experimental structure prediction). The manuscript is suitable for publication in Nature Communications after the following points are addressed.

Thank you for your thoughtful and encouraging assessment of our work.

1-On line 263 the authors state "Dcp1 does not directly stimulate Dcp2 in either species but mediates stimulation by decapping enhances." This statement is not entirely supported by their data and the published literature. Yeast Dcp1 was shown to stimulate Dcp2 by around 10-fold (PMID:22323607). I doubt this stimulation could be observed in the kinetic analyses reported in Figure 5f due to the paucity of data in the initial phase of the exponential. To make this claim, the authors should accurately quantify decapping rates by recording a handful of timepoints in the early phase of the exponential (between 0 and 5 minutes). In addition, the authors should take care that the same decapping buffer is employed with prior studies if comparisons are to be made. In particular, Mg^{2+} is sufficient for decapping by yeast Dcp1/Dcp2; the addition of Mn^{2+} to the buffer will accelerate the rate of the chemical step such that the fraction of the closed form of the enzyme is no longer rate-limiting, potentially masking the effect of Dcp1. Alternatively, the authors should tone down their claim that yeast Dcp1 does not affect decapping by Dcp2.

We thank the reviewer for the careful reading of our manuscript and for pointing out the discrepancy with earlier work on *S. pombe* Dcp1/Dcp2 activation (PMID: 22323607). We have conducted the additional experiments as suggested and have revised both the text and figures accordingly.

To address the question of whether the presence of Mn^{2+} was masking the previously reported stimulation of *Sp* Dcp2 by *Sp* Dcp1, we conducted additional decapping assays in the presence and absence of Mn^{2+} . In agreement with previous work, we observed that Mn^{2+} does stimulate decapping by *Sp* Dcp2/*Sp* Dcp1 when directly compared with reactions containing only Mg^{2+} .

Additionally, we made the following alterations to the in vitro decapping assay.

- **Additional early time points.** To capture the initial phase of the reaction progress curve, reactions were quenched at 0 min, 2.5 min, 5 min, 10 min, 20 min, and 40 min (replacing the former 0/10/20/40 min schedule). It should be noted that we were unable to obtain consistent time point sampling at the very beginning of the reaction (0.5-2.5 min).
- **Lower Dcp2 concentration (100 nM).** This shifted the reaction into the linear range, minimizing substrate depletion.
- **Dcp1:Dcp2 ratios of 1:1 and 10:1.** These match the conditions previously described (PMID: 18280238).

In summary, Mn^{2+} accelerated cap hydrolysis by the *Sp* Dcp2(1-242)/*Sp* Dcp1 complex, when directly compared with reactions containing only Mg^{2+} , but higher Mn^{2+} -driven baseline activity did **not** obscure the magnitude of Dcp1-dependent stimulation. Therefore, we conclude that Mn^{2+} enhances the intrinsic catalytic step of Dcp2, whereas Dcp1 contributes little additional activation under either metal condition under the assay conditions tested in our study.

Manuscript updates:

- **Results** – incorporated the new data and addressed the additional validation of the effects of Dcp1 on Dcp2 in different reaction regimens.
- **Supplementary Figure 5** – new panels with expanded early time points, different Dcp1:Dcp2 ratios, and Mg^{2+} vs Mn^{2+} comparisons.

- **Methods** – added more information on buffer compositions used.
- **Discussion** – commented on the lack of activation of *Sp* Dcp2 by *Sp* Dcp1 in our assays under different conditions.

These additional experiments confirm that, in our hands, *Sp* Dcp1 confers only a limited activation of *Sp* Dcp2, and this conclusion is now presented with the appropriate nuance. Notably, earlier kinetic analyses were performed at considerably higher protein concentrations (1-10 μM *Sp*Dcp2/*Sp*Dcp1) than those used here (0.1 μM), a difference that likely contributes to lab-to-lab variability in the magnitude of Dcp1 stimulation. We appreciate the reviewer's guidance, which has strengthened both the data and the clarity of our argument.

2-The authors should clarify which experiments were performed without MBP tags in the Methods section. I could only find the details of tag removal for hDcp1.

We appreciate the reviewer's attention to this detail. We now explicitly state that **all proteins retained their N-terminal MBP tags, unless otherwise noted**. The sole exception is **human DCP1 (1-135)**, from which the MBP tag was proteolytically cleaved before use. This addition should make it unambiguous which constructs were used in their tag-free form.

Manuscript updates:

- **Methods** – clarified which constructs were left fused to MBP and in which the tag was removed by protease cleavage.

3-The authors may wish to mention the tumor suppressor PNRC1 has a DAM and probably acts similarly to PNRC2 (PMID: 30373810).

Thank you for pointing out the relevance of PNRC1. We now note that the tumor suppressor PNRC1 also harbors a Decapping Activator Motif (DAM) and, based on structural and functional parallels (PMID: 30373810), is likely to promote DCP1/DCP2 activation in a manner analogous to PNRC2.

Manuscript updates:

- **Results** – mentioned that PNRC1 and PNRC2 both contain a DAM and cited Gaviraghi et al.
- **Discussion** – drew an analogy between PNRC1 and PNRC2 in decapping activation.

These additions highlight the broader relevance of DAM-containing proteins to decapping regulation without altering our main conclusions.

4-Edc4 has a WD40 domain. Why was this omitted from the alpha-fold structure prediction?

At the time of writing, we are constrained by GPU memory on our institutional computing cluster. The human EDC4 tetramer (4 × 1,401 amino acids) **exceeded the practical sequence-length limit for structure predictions**, so we predicted only the coiled-coil segment.

However, since then, we have generated AlphaFold-Multimer predictions of the complete *C. elegans* EDC4 tetramer (4 × 841 amino acids), which now fully resolve the WD40 domain and its orientation relative to the coiled-coil. The resulting model of the full-length protein agrees well with the EM class

averages in the coiled-coil region and places the WD40 domains radially around the tetrameric core. We note, however, that given the linker length between the helical portion and the WD40, the arrangement is likely completely stochastic.

Manuscript updates:

- **Supplementary Figure 7c** – structure prediction of full-length *C. elegans* EDC4 tetramer.
- **Results** – described the full-length *C. elegans* EDC4 tetramer structure prediction.

We hope this addresses the reviewer's concern and provides a more complete picture of EDC4 architecture.

Reviewer #2 (Remarks to the Author):

Much of the field's understanding of mRNA decapping is derived from studies of the yeast Dcp2 decapping protein and its activity/stimulation. Relatively less is known in metazoan systems. This manuscript uses a reconstituted system to characterize the decapping by human DCP2 decapping protein and stimulation of its activity. Importantly, they show that unlike yeast Dcp2, the decapping stimulator DCP1 does not directly interact with or stimulate DCP2 in humans. Interestingly, they show the interaction and stimulation is mediated by a third protein, PNRC2 and further characterize the stoichiometry of the protein complex. The difference between the metazoan and yeast systems is an important addition to the field. The authors also study the metazoan restricted scaffold protein, enhancer of decapping 4 (EDC4) and its structure and activity. Contrary to previous reports, they show that the EDC4 does not stimulate DCP2 decapping. The data are sound and the findings interesting. Several points are detailed below to enhance the manuscript.

Thank you for your enthusiasm for our work and thoughtful feedback. We have carefully addressed your detailed suggestions to improve the manuscript.

1. EDC4 has long been thought to serve as a scaffold that juxtaposes decapping factors to stimulate decapping. In Sup Fig 6, the authors show EDC4 does not stimulate DCP2 decapping and surprisingly inhibits it. They further show the EDC4 also inhibits decapping when DCP1 was included in the reaction. However, considering their findings that DCP1 does not interact with DCP2 and requires PNRC2 to stimulate Dcp2 decapping, this later finding is not surprising. These studies should also be done with the inclusion of PNRC2 to test whether EDC4 can further enhance the stimulation of DCP1/PNRC2 on DCP2 decapping.

We are grateful for the reviewer's suggestion to test whether EDC4 can potentiate the DCP1/PNRC2-mediated stimulation of DCP2. We have performed the requested experiments and incorporated the results into the revised manuscript.

- **Protein concentrations:** Reduced to 50 nM to maximize sensitivity to additional stimulation.
- **Early time points:** A 2.5-minute quench was added (0, 2.5, 5, 10, 20 minutes) to resolve the initial exponential phase.
- **Combinations tested:** *Hs* DCP2 alone, *Hs* DCP2 + *Hs* EDC4, *Hs* DCP2 + *Hs* DCP1 + *Hs* EDC4, *Hs* DCP2 + *Hs* DCP1 + *Hs* PNRC2, and *Hs* DCP2 + *Hs* DCP1 + *Hs* PNRC2 + *Hs* EDC4.

We observed that, under conditions where PNRC2 robustly stimulates DCP2/DCP1, EDC4 does not provide additional catalytic enhancement. This finding refines the prevailing model of EDC4 as a scaffold. While it may organize decapping components *in vivo*, it does not accelerate the chemical step of decapping in our reconstituted system. We thank the reviewer for prompting this informative set of experiments, which has strengthened the manuscript.

Manuscript updates:

- **Results** – added description of the results of the PNRC2 ± EDC4 assays.
- **Supplementary Figure 8c,d** – decapping assays with EDC4 and PNRC2.
- **Discussion** – stated that EDC4 does not enhance catalysis even when PNRC2 is present, implying its primary role is likely architectural or regulatory rather than catalytic.

2. Markers on the TLC of Fig 2 should be included. Are the products m⁷GMP or m⁷GDP? Why does the product size shift over time?

We thank the reviewer for pointing out the need for clear TLC markers and for asking about the identity and apparent mobility change of the reaction product. Each TLC of every figure has been labelled on the right with a filled circle denoting the position of the **origin/substrate** and an empty circle for the position of the **m⁷GDP product**. Authentic radiolabelled standards previously run in parallel confirm that the sole product comigrates with **m⁷GDP**, not m⁷GMP. This is also as expected for DCP2, which cleaves the β-γ phosphoanhydride bond to release m⁷GDP.

Manuscript updates:

- **Methods** – added information on TLC markers.
- **Figures 2/Supplementary Figures 4,5,6,8** – markers indicated and explicitly mentioned in the legends.

The slight downward displacement of the product spot at later time points is an artifact of TLC plate positioning. Minor differences in the insertion depth of each plate (±1–2 mm) altered the total solvent-front migration, resulting in a slight apparent offset in band position when the plates were scanned together.

Manuscript updates:

- **Methods** – added text to describe the differences in solvent front migration.

We hope these additions fully address the reviewer's concerns about TLC annotation and product identification.

3. Do the author have an explanation for why the DCP2 1-245 fragment does not bind RNA (fig 3b), but can still decap RNA (fig 2c)?

We agree that at first glance it seems paradoxical that the *Sp* Dcp2 (1-245) fragment fails to show detectable RNA binding in our electrophoretic mobility-shift assay (EMSA, Fig. 3b) yet is catalytically competent in the decapping assay (Fig. 2c). Two points reconcile these observations:

1. **Different regions of Dcp2 drive cap recognition versus “bulk” RNA binding.** Residues 1–245 encompass the entire catalytic Nudix fold and the immediately adjacent regulatory domain. Structural work on *S. pombe* and human decapping complexes (PMID: 28533364, 27694842) shows that virtually all direct contacts with the cap nucleotide, including those that position the methylguanine and the triphosphate for hydrolysis, reside within this N-terminal segment. In contrast, this study demonstrates that **high-affinity interactions with the RNA body are localized to the C-terminal intrinsically disordered region of the protein**. This stretch likely provides a positively charged “noodle” that wraps around the transcript and stabilizes the open, RNA-bound conformation. Truncating at 245, therefore, removes most of the electrostatic contacts that produce a discrete, slowly exchanging complex on an EMSA gel.
2. **Catalysis relies on transient cap engagement, not on tight binding to the RNA body.** In the decapping assay, the substrate concentration (100 nM) is close to the enzyme concentration, so even short-lived collisions that align the cap within the Nudix active site are sufficient for cleavage to occur. The EMSA, by design, detects only those complexes that survive gel electrophoresis (~60 min), which requires K_d in the low-nanomolar range. If the K_d of Dcp2 (1-245) for the full transcript is $\geq 0.5 \mu\text{M}$, within the range reported for isolated Nudix cores (PMID: 29618050), the complex will dissociate during the run and appear “non-binding” while still turning over the substrate in solution.

Manuscript updates:

- **Results** – clarified why RNA binding and catalytic activity do not correlate directly for the C-terminally truncated DCP2 construct.
4. The identity of the various Short Functional Domains shown as black bars in Fig 1 should be defined. The phenylalanine-rich EDC4 binding (FEB) motif should also be defined.

Thank you for requesting a clearer annotation of the short functional domains in Fig. 1. We have revised the figure and legend accordingly.

The black bars are now labeled with their accepted name (**HLM** – Helical Leucine-rich Motif) and residue ranges. **FEB (phenylalanine-rich EDC4-binding) motif** is now identified as the C-terminal region in DCP2 (residues 395-420), which contains critical aromatic residues required for the direct interaction with EDC4 (PMID: 19966221). The FEB motif is now shown in dark gray and annotated directly on the schematic.

Manuscript updates:

- **Figure 1** – the legend now provides a complete description for each motif.

Reviewer #3 (Remarks to the Author):

The publication by Simko & Muthukumar, Valcov describes a noteworthy effort to reconstruct human decapping mechanism in vitro using full length recombinant proteins. For many years the decapping reaction was extensively studied in vitro and in vivo using the budding and fission yeast models. Most

key actors in this process are conserved or have identified counterparts in human. However, the considerable difference in amino acid sequence and the presence of additional domains in human proteins, suggests that the mechanistic details of human decapping might be very different from the yeast system. Therefore the work by Simko & Muthukumar, Valcov is a much needed step towards our full understanding of the decapping process.

The Introduction is well structured, supplemented with schemes, and clearly puts forward the known functions and mechanism of action of yeast decapping enzyme and its co-factors altogether putting forward the limited differences readily identifiable from evolutionary amino-acid sequence changes. Using a mix of approaches including in vitro assays using human recombinant proteins, protein interaction modeling and structural or semi-structural studies the Result section systematically characterizes the decapping reaction in vitro. The Discussion clearly sums up the in vitro results, that show a rearrangement of molecular functions in the human system compared to the yeast model. It also puts the multimerization data in the in vivo context of in vivo P-body formation; an aspect which is a bit more far fetched, but still acceptable in this section.

Altogether, the publication is engagingly written, filled with interesting data, and will be a good addition to Nature Communication portfolio. Given the rich collection of mechanistic data gathered in yeast, a systematic comparison to the human system is much needed in the decapping context, and beyond. I would therefore recommend the work for publication after the authors answer the few questions below.

Thank you for your thoughtful and encouraging review. We greatly appreciate your recognition of the value of comparative biochemical analyses and have used your insights to further refine and strengthen the manuscript.

Major:

Question 1:

Fig S1 - please present more clearly the hsDcp2 amino-acids presumably involved in RNA binding (Fig. S1f - number of aa type of aa), and if possible attempt to validate them by mutational analysis. What types of RNA sequences were used for modeling? Are those 5'UTRs of actual mRNAs? Was a cap included?

We appreciate the reviewer's interest in a more precise depiction of the DCP2 C-terminal residues predicted to engage RNA and in potential experimental validation.

Predicted RNA-binding residues. We mapped the fIDPnn per-residue scores onto the C-terminal low-complexity segment, encompassing all residues (residues 242-420) in a Jalview-style representation. Residues are individually color-ramped from white (score < 0.3) to dark red (score \geq 0.8), with the score scale shown alongside.

Supplementary Figure 1f is meant to convey the **overall clustering** of positively charged residues that fIDPnn predicts to populate the intrinsically disordered C-terminal tail (residues 242-420). It is **not intended to pinpoint atom-level contacts**, whose accuracy would be poor given the limited resolution

of current disorder-to-structure prediction tools for protein-RNA interactions. We have clarified this in the figure legend.

Manuscript updates:

- **Supplementary Figure 1b** – fIDPnn per-residue scores of predicted RNA-binding propensity have now been mapped explicitly on the whole sequence of DCP2 C-terminal residues.
- **Supplementary Figure 1f** – figure legend modified to state: “It should be noted that the cartoon illustrates the spatial clustering of positively charged residues inferred from modeling; individual residue contacts are not defined.”

RNA substrate. To be consistent with previous in vitro work on metazoan decapping factors (PMID: 24510189, 19966221) and to ensure that the modeled RNA matches the substrate used in our biochemical assays, we employed the same RNA fragment derived from the *Drosophila melanogaster* Me31b-encoding transcript (the sequence is provided in the **Supplementary Table 1**). The RNA was modeled **without** an m⁷G cap (due to the technical limitations of the software); therefore, the prediction focuses exclusively on protein contacts with the RNA body.

Manuscript updates:

- **Methods** – added the information on the RNA substrate used for modeling.

Feasibility of systematic mutational analysis. The predicted RNA interface spans 180 residues, dispersed throughout a highly charged, low-complexity region. Creating the >20 single-alanine and/or reversed-charge variants needed to test the highest-scoring residues, and then expressing and purifying each mutant to test it in an assay with labeled isotopes, would exceed the scope of the current study. We cannot, frankly, do this given our current institutional constraints. We agree that a comprehensive scanning mutagenesis campaign will be valuable, but it warrants a dedicated follow-up study.

Question 2:

Section: ,Trimerization Dynamics and Interaction Specificity of Human DCP1' - when preparing recombinant proteins for in vitro decapping experiments the authors performed size exclusion chromatography (as stated in M&M section). Was the multimerization visible using this technique? An orthogonal approach is always welcome in addition to mass photometry.

Recombinant proteins like to create aggregates, thus I personally would downplay the conclusion about any higher (than a trimer) order structures until a clear motif responsible for their formation can be isolated. Even if this is highly interesting in the context of Dcp participation in P-bodies.

We thank the reviewer for highlighting the value of an orthogonal method to complement our mass photometry (MP) analysis and for cautioning regarding higher-order assemblies. We have now included the analytical size exclusion chromatography (SEC) profiles of full-length wild-type DCP1 and the trimerization domain mutant (TDm).

Manuscript updates:

- **Supplementary Figure 2** – analytical size exclusion chromatography (SEC) profiles of full-length wild-type DCP1 and the trimerization domain mutant (TDm).
- **Results** – now describe the SEC observations with DCP1 constructs.

We agree that recombinant IDR-rich proteins can aggregate and that trimerization alone does not imply the existence of larger, physiologically relevant oligomers. We left the text that simply stated that larger species were observed and deleted this overly speculative sentence *“These DCP1-TDm species indicate that homomeric association occurs beyond the known TD interface, revealing a novel mode of interaction which could contribute to the oligomerization of trimers into higher-order assemblies.”*

Analytical SEC corroborates our mass photometry findings and pinpoints the C-terminal motif as the structural determinant of DCP1 trimerization. We have moderated statements about larger oligomers until a specific interface for such assemblies can be demonstrated. We appreciate the reviewer’s constructive suggestions, which have strengthened both the experimental support and the balance of our conclusions.

Best wishes in the New Year!

Thank you!

AUTHOR RESPONSE TO REVIEWERS

We thank the reviewers for their additional comments, which have helped us improve the manuscript. All textual changes are highlighted in red in the revised manuscript, and our point-by-point responses are provided below in blue for ease of navigation.

Reviewer #1 (Remarks to the Author):

The revised manuscript by Simko addresses all of my questions except the one about the inconsistencies between the stimulatory effects of Dcp1 on Dcp2's catalytic activity reported in previous studies and their manuscript.

I appreciate their testing of the role of metal ions in decapping reactions; however, upon initial review, it was unclear whether decapping experiments were performed with MBP tags attached to Dcp2. The authors clarified in the revised manuscript that MBP tags were retained on the tested Dcp2 constructs. Therefore, the statement "Our findings reveal a divergence in the function of the DCP2 C-terminal sequence between yeast and human homologs, with no direct stimulation of DCP2 by DCP1 observed" on lines 86-87 is not supported by well-controlled experiments, as they have not ruled out the possibility that the latter result is an artifact of using MBP-tagged Dcp2. Indeed, prior studies showing a stimulatory effect of spDcp1 were performed with untagged spDcp2 (PMID:20711189; 22323607), and this conclusion is supported by a study from Valkov et al., which showed spDcp1 could stimulate RNA binding and decapping by spDcp2 (PMID:27183195, Supplemental Figure 4e, 4h). Notably, the Dcp1 and Dcp2 constructs used in the decapping assays reported by Valkov et al did not contain MBP or any other protein tags (PMID:27183195, Supplemental Figure 4i).

Thus, it seems likely that a source of discrepancy regarding Dcp1 stimulation of decapping by Dcp2 could be the retention of the MBP tag on Dcp2. This possibility should be investigated using EMSA and/or decapping assays with constructs that lack MBP tags. While Dcp1 can bind MBP-tagged Dcp2, this does not rule out the possibility that the MBP tag might interfere with RNA binding or catalytic enhancement, especially considering that this system undergoes conformational changes during catalysis (PMID:28533364).

Overall, this manuscript presents important new insights into the human decapping complex. I agree with the authors' assertion that the conserved function of Dcp1 is to recruit Edc1-type coactivators to stimulate Dcp2 catalysis. However, their claim that Dcp1 does not stimulate decapping in the absence of Edc1 or PNRC2 is based on experiments with MBP-tagged Dcp2 constructs, which could confound interpretation. The manuscript is worthy of publication, but the discrepancy with prior studies should be addressed either by additional experiments or by qualifying their conclusion about Dcp1's role to state 'the effect of the MBP tag on Dcp2's ability to be stimulated by Dcp1 cannot be ruled out' in the Discussion.

We agree that it is important to be transparent about the potential influence of the MBP tag on biochemical behavior. In our hands, retaining the N-terminal MBP tag on human DCP2 is a practical necessity for biochemical tractability: it substantially improves soluble expression, yield, and sample homogeneity, and enables preparation of material suitable for quantitative decapping assays. Untagged (or minimally tagged) full-length human DCP2 constructs were not comparably tractable under our

purification and assay conditions, limiting our ability to perform the full set of controlled comparisons in a consistent manner.

We also agree that, because our key measurements were obtained with MBP–DCP2, we cannot formally exclude the possibility that MBP influences the magnitude of any Edc1/PNRC2-independent contribution of DCP1. Accordingly, we have added the following sentence to the **Discussion** to qualify this conclusion: “However, because we used MBP-tagged constructs for biochemical tractability, we cannot rule out that the MBP tag alters DCP2’s responsiveness to DCP1.”

This change addresses the discrepancy with prior studies without overstating mechanistic conclusions beyond what is directly supported by the constructs that are experimentally tractable in our system.

Minor:

1. The authors' explanation in the rebuttal that the inconsistency is due to 'lab to lab variation' is not correct. They mention, “Notably, earlier kinetic analyses were performed at considerably higher protein concentrations (1-10 μM SpDcp2/SpDcp1) than those used here (0.1 μM), a difference that likely contributes to lab-to-lab variability in the magnitude of Dcp1 stimulation.” The effect of Dcp1 should be evident in the observed rate (k_{obs}) measured under single-turnover conditions, where k_{obs} is proportional to k_{max}/K_m or k_{max} depending on whether the decapping enzyme concentration is below or above K_m , at low (100 nM) or high (10 μM) enzyme concentrations.

We appreciate the reviewer’s clarification regarding single-turnover kinetics and agree that our prior wording in the rebuttal (“lab-to-lab variation”) was imprecise. Under single-turnover conditions, K_{obs} depends on enzyme concentration according to $k_{\text{obs}} = k_{\text{max}}[E]/(K_m + [E])$. Consequently, the apparent magnitude of Dcp1 stimulation can differ between assays performed at 0.1 μM versus 1–10 μM Dcp2, depending on which kinetic parameter Dcp1 primarily affects. If Dcp1 increases k_{max} , stimulation should remain evident even when $[E] \gg K_m$. In contrast, if Dcp1 mainly decreases the apparent K_m (or increases the fraction of productive enzyme-substrate complexes), the effect will be most pronounced when $[E] \ll K_m$ and reduced as $[E]$ approaches or exceeds K_m .

In our assays, the total capped RNA substrate was ~40 nM (assuming 100% labeling efficiency), while Dcp2 was typically 200 nM, i.e., the enzyme was in ~5-fold excess over substrate. We have added a sentence to the **Methods** to explicitly state the RNA concentration used in the decapping assays. Although our design is consistent with pseudo-first-order (single-turnover) conditions with respect to enzyme:substrate stoichiometry, our enzyme concentration remains substantially lower than the 1–10 μM used in earlier studies. Thus, if prior work were closer to saturation with respect to K_m , whereas our conditions are sub-saturating, the quantitative fold stimulation by Dcp1 would not be expected to match across studies even when k_{obs} is extracted under single-turnover analysis.

2. There is no section in the Methods describing how spDcp2 (1-242) was expressed and purified.

The expression and purification of Sp Dcp2(1–242) is described in the **Methods** in the paragraph beginning “Sp Dcp2 including a portion of C-terminal IDR sequence ...”, where we state: “Sp Dcp2(1–242) was purified using the same procedure as Sp Dcp2(1–504) with the concentration step omitted.”

Reviewer #2 (Remarks to the Author):

The authors have adequately addressed my initial concerns, and I would recommend publication.

Thank you.

Reviewer #3 (Remarks to the Author):

The revised work by Simo et al. is a purely in vitro study brings forward the evolutionary reshuffling of functions in the decapping machinery protein domains (DCP1/2, EDC4, PNRC2), mainly between the human and *S. pombe* systems. Additionally, the work provides experimental structural information about the EDC4 C-terminal coiled-coil domain, which mediates the protein's multimerization. Overall, the study gives much in vitro information to inspire follow up in vivo studies. This I hope will be a much desired outcome of this work.

the authors have answered my previous questions and I think the publication should be accepted, with minor corrections to figures listed below. Those can be added in proof.

Minor:

Some of the introductory comparative experiments were performed with large fragments of *S. pombe* Dcp2, in contrast to the use of full length human protein. The *S. pombe* protein is very large, which was likely thereason for its fragmentation. The decision and the origin of the construct should be better explained in the first paragraph of the results.

We now clarify this explicitly in the **Results** as follows: "Because *S. pombe* Dcp2 is considerably larger and contains extensive low-complexity regions that limit recombinant expression and sample homogeneity, we performed cross-species comparisons using a large *S. pombe* Dcp2 fragment encompassing the conserved catalytic core and a segment of the C-terminal low-complexity region (residues 1–504), which retains decapping activity in vitro and has been used previously for biochemical studies (Paquette et al. 2018); full-length human DCP2 is comparatively tractable and was therefore used throughout."

The initial comparative experiments were designed to benchmark key biochemical behaviors of Dcp2 across species, using constructs that are technically tractable for recombinant production/purification, and subsequent experiments. Full-length *S. pombe* Dcp2 (741 residues) is substantially larger than the human enzyme (420 residues). It contains extended low-complexity/disordered regions, which, in our hands, reduced expression yields and sample homogeneity for the full-length protein. We therefore used a previously validated (PMID: 29618050) large *S. pombe* Dcp2 fragment that encompasses the conserved N-terminal regulatory domain (NRD) and Nudix catalytic domain (i.e., the structured core responsible for catalysis and most conserved regulatory interactions), as well as a substantial portion of the unstructured C-terminal region while omitting distal low-complexity extensions that are dispensable for basal in vitro decapping activity but compromise biochemical handling.

Fig. 1 - add numbers indicating length of depicted proteins and position of highlighted domains.

Done.

Fig. 4 - is the trimer 187 or 188 kDa? there is a discrepancy between fig. 4a and 4b? A 59 kDa monomer adds up to a 177 trimer. This whole panel can be a bit confusing. I understand that the method gives a mol. weight range. Maybe indicate the range of monomer and trimer size estimates in a smaller and grey font so that it becomes less confusing? The peak estimate can be left as it is.

It is important to note that mass photometry reports an *apparent* molecular mass inferred from interferometric scattering contrast relative to calibration standards. Minor day-to-day differences in calibration, acquisition (as well as distribution broadness), and histogram binning can shift the reported peak position by ~1–2 kDa without changing the underlying interpretation.

Regarding the point that $3 \times 59 \text{ kDa} = 177 \text{ kDa}$, we note that both 59 kDa (monomer) and 187–188 kDa (trimer) are *apparent* mass estimates obtained from separate distributions and are subject to the intrinsic uncertainty of the method and its calibration with predominantly globular standards. A case in point is that the actual molecular weight of the DCP1 monomer, based on its sequence, is 63.3 kDa (and a trimer is therefore $3 \times 63 \text{ kDa} = 189 \text{ kDa}$, which is in close agreement with the estimated masses here). DCP1 constructs, which contain extensive low-complexity/disordered regions that can deviate from the behavior of the globular proteins used for calibration of the mass photometer. The key conclusion is unchanged: wild-type DCP1 is dominated by a trimeric population, whereas the TD mutant shifts strongly toward a monomeric population.

We considered adding explicit “ranges” to the plot, but chose not to because (i) it would materially increase visual complexity, and (ii) it risks implying a level of precision/interpretability that is not warranted for these broad, heterogeneous distributions. Instead, we have clarified in the figure legend that the annotated values correspond to the peak positions of the mass photometry distributions (apparent masses) and may differ slightly across measurements due to calibration and peak-picking.

Fig. 5d - the identity of the regions highlighted should be indicated as in fig. 5c

Done.

Fig. S1A - scale missing on x-axis (number of amino acids). This will enable to better compare panel a with panel b

Done.

Fig. S2 - please include a molecular weight scale for the SEC analysis (can be from the column's manual) in parallel to the elution volume. This will help to verify if the monomer size corresponds to the Mol. weight of DCP1.

Done.